



# Quality-controlled meteorological datasets from SIGMA automatic weather stations in northwest Greenland, 2012–2020

Motoshi Nishimura[1*], Teruo Aoki[1], Masashi Niwano[2], Sumito Matoba[3], Tomonori Tanikawa[2], Tetsuhide Yamasaki[4], Satoru Yamaguchi[5], Koji Fujita[6]

[1]National Institute of Polar Research, Tokyo, Japan

[2]Meteorological Research Institute, Japan Meteorological Agency, Ibaraki, Japan

[3]Institute of Low Temperature Science, Hokkaido University, Hokkaido, Japan

[4]Avangnaq Arctic Project, Osaka, Japan

[5]Snow and Ice Research Center, National Research Institute for Earth Science and Disaster Resilience, Niigata, Japan

[6]Graduate School of Environmental Studies, Nagoya University, Nagoya, Japan

*Correspondence to:* Motoshi Nishimura (nishimura.motoshi@nipr.ac.jp)

**Abstract.** In situ meteorological data are essential to better understand ongoing environmental changes in the Arctic. Here, we present a dataset of quality-controlled meteorological observations by two automatic weather stations in northwest Greenland from July 2012 to the end of August 2020. The stations were installed in an accumulation area on the Greenland Ice Sheet (SIGMA-A site, 1490 m a.s.l.) and near the equilibrium line of the Qaanaaq Ice Cap (SIGMA-B site, 944 m a.s.l.). We describe the two-step sequence of quality-control procedures that we used to create increasingly reliable datasets by masking erroneous data records. We analyzed the resulting 2012–2020 time series of air temperature, positive degree-days, snow height, surface albedo, and histograms of longwave radiation (a proxy of cloud formation frequency). We found that snow height increased and albedo remained steady at the SIGMA-A site, whereas high air temperatures and clear-sky conditions prevailed while snow height and albedo decreased in the summers of 2015, 2019, and 2020 at the SIGMA-B site. Therefore, it appears that these weather conditions led to notable snow height degradation at the SIGMA-B site but not at the SIGMA-A site. We anticipate that this quality-control method and these datasets will aid in climate studies of northwest Greenland as well as contribute to the advancement of broader polar climate studies.

## 1. Introduction

Recent changes of the Greenland Ice Sheet have likely contributed to the global rise in sea level



(e.g., IPCC, 2021). These changes include rising air temperature on the ice sheet, the increasing extent
of bare and dark ice (Shimada et al., 2016), and the loss of ice mass (Hanna et al., 2013; IMBIE Team,
2020). Many studies have used regional climate models and atmospheric reanalysis data (e.g., Niwano
et al., 2018; Fettweis et al., 2020) to reveal major ablation events in Greenland and to reconstruct the
long-term past surface mass balance of the Greenland Ice Sheet. In situ meteorological data provide
vital information to monitor environmental changes and inform the models that simulate them;
however, the existing in situ meteorological data are insufficient for these purposes.
Some automatic weather station (AWS) networks have been constructed on the Greenland Ice
Sheet, including GC-Net (Steffen and Box, 2001) and PROMICE (van As et al., 2011; Fausto et al.,
2021), and have provided important long-term meteorological data. To contribute to these efforts and
to fill a spatial gap, we established two AWS systems in northwest Greenland (Fig. 1), where rapid
environmental changes have occurred in recent years (Aoki et al., 2014). Recent studies of this region
have documented a drastic mass loss since the mid-2000s (Mouginot et al., 2019), an expansion of the
ablation area (Noël et al., 2019), and a hot spot of increasing rainfall (Niwano et al., 2021). The two
sites were established in 2012 as a part of the Snow Impurity and Glacial Microbe effects on abrupt
warming in the Arctic (SIGMA) Project, which aimed to clarify the dramatic enhancement of melting
of the Greenland Ice Sheet induced by snow impurities (e.g., black carbon, mineral dust). The
observational data acquired since that time have been used by glaciological (Yamaguchi et al., 2014;
Tsutaki et al., 2017; Matoba et al., 2018; Kurosaki et al., 2020), meteorological (Aoki et al., 2014;
Tanikawa et al., 2014; Niwano et al., 2015; Hirose et al., 2021), and biological studies (Onuma et al.,
2018; Takeuchi et al., 2018). These data are also valuable because they support the analytical values
of various numerical models (e.g., Niwano et al., 2018; Fujita et al., 2021) and form the basis for
robust analytical results.
The datasets from AWS generally contain erroneous data records that are attributed to sensor noise
or natural factors. Various procedures exist for improving the accuracy of such datasets (e.g., Fiebrich
et al., 2010; Fausto et al., 2021). In particular, careful QC procedures are required for downward
radiation sensors, which are sensitive to solar zenith angle, icing, riming, and snowfall (van den Broeke
et al., 2004a, b; Moradi, 2009). Other QC procedures deal with error sources through range, step, and
internal consistency tests (Estévez et al., 2011). The specifics of QC methods, for example, the
threshold value for detecting erroneous data records, should be adjusted for each observation
environment. In this paper, we describe the QC methods used for the in situ meteorological observation
data from northwest Greenland, which include existing QC methods, new ones, and combinations of
both.
After describing the AWS sites (Sect. 2) and their datasets (Sect. 3), this paper introduces the two
separate QC methods used sequentially to mask erroneous data records (Sect. 4). We then present
examples of time series of meteorological variables in northwest Greenland, infer their implications





for interannual variations in weather conditions, and describe the differences between the two sites
(Sect. 5).
**2. Site description**

The two AWSs are installed at the SIGMA-A site (78.052° N, 67.628° W; 1490 m a.s.l.), on the
northwest Greenland Ice Sheet, and the SIGMA-B site (77.518° N, 69.062° W; 944 m a.s.l.), on the
Qaanaaq Ice Cap, a peripheral ice cap on the Greenland coast (Fig. 1). They have been in operation
since July 2012 (Aoki et al., 2014).
The SIGMA-A site is 70 km inland from the coast on a ridge of the Greenland Ice Sheet extending
northwest from the Greenland Summit; it sits on a flat snow surface with no obstacles around the site
(see Fig. 2). It is considered to be in an accumulation area for the ice sheet (Matoba et al., 2018) based
on the analysis of ice-core data (Yamaguchi et al., 2014; Matoba et al., 2017). The SIGMA-B site is 3
km north of the village of Qaanaaq. Its location is supposed to be near the equilibrium line (910 m
a.s.l.; Tsutaki et al., 2017) on the Qaanaaq Ice Cap, which ranges in elevation between 30 and 1110 m
a.s.l. (Sugiyama et al., 2014). The surface condition at this site varies (see Fig. 2), and surface melting
has occurred in warm years (e.g., Aoki et al., 2014). The site is on a southwest-facing slope (azimuth
220°) with an angle of 4° according to 10 m DEM data (Porter et al., 2018).

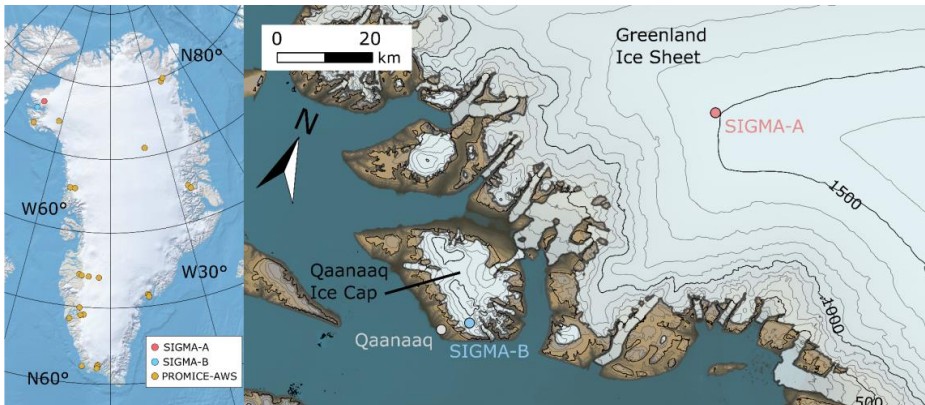

Figure 1. Location map of Greenland showing AWS sites (left) and a local map of northwest Greenland
showing locations of AWS sites SIGMA-A and SIGMA-B. Contour interval in the right panel is 100
m.



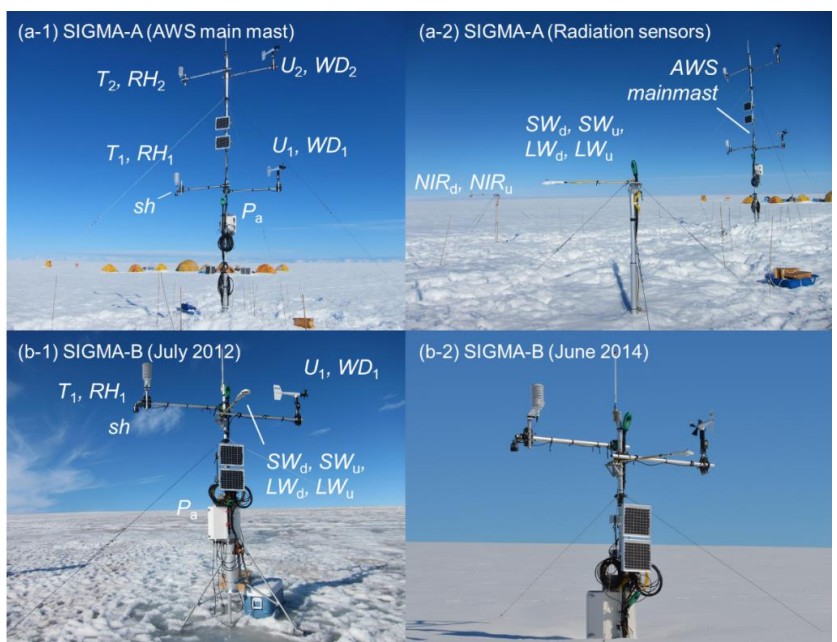


Figure 2. Setting and instrumentation at the SIGMA-A site (top) and the SIGMA-B site (bottom).
Surface conditions at SIGMA-B are shown in July 2012 and June 2014. Sensors are labeled with the
observation parameters they measure (see Table 1).

## 3. Description of AWS systems and datasets

### 3.1. Specifications

Sensor specifications for the meteorological observations are listed in Table 1, and overviews of
the two AWS systems are shown in Fig. 2. Each AWS mainmast is set in a hole drilled using a hand
auger. Sensors for air temperature, relative humidity, and wind speed and direction are mounted at the
ends of horizontal poles to exclude possible thermal and wind disturbances from the mainmast. The
SIGMA-A sensors are placed 3 m and 6 m above the surface, as signified by subscripts "1" (lower)
and "2" (upper) in the corresponding data variables. The SIGMA-B sensors are set at 3 m above the
surface and have subscripts of "1". The snow height sensor at both sites is mounted at 3 m height
beneath the air temperature and relative humidity sensors. Six snow temperature sensors have been set
as follows. Four sensors were set at 19:00 UTC on 29 June 2012 at depths of 1 m ($st_1$), 0.7 m ($st_2$), 0.4
m ($st_3$), and 0.05 m ($st_4$) under the snow surface. At 21:00 UTC on 27 July 2013, sensors $st_3$ and $st_4$
were relocated to depths of 0.46 m and 0.16 m, respectively. Sensors $st_5$ and $st_6$ were set at 0.05 m



under the surface and 0.45 m above the surface, respectively, at 14:00 UTC on 9 June 2014. Sensors
for shortwave, longwave, and near-infrared radiation are installed at SIGMA-A on separate poles 10
m from the mainmast (Fig. 2a-2). A pyranometer and a pyrgeometer at SIGMA-B are mounted on the
mainmast facing directly south. Tilt angles of the mainmast in the north-south ($Tilt_X$) and east-west
($Tilt_Y$) directions are monitored with an inclinometer attached to the mainmast. The additional suffix
"A" or "B" represents the site name in the variables introduced below.
Electric power is supplied to the AWS systems by a cyclone battery that is charged constantly by
solar panels attached to the mainmast. All parameters are recorded once per minute and stored in a
data logger (C-CR1000, Campbell Scientific, USA), except for the mainmast's snow height and tilt
angles, which are recorded every hour. Hourly data are calculated for the other parameters by
averaging the 1-min data. All hourly data are sent regularly to the data server via the Argos satellite
channel.
Snow height is measured with an ultrasonic snow gauge (Table 1). The raw data from this sensor
($sh_{raw}$) is the distance from the sensor to the snow surface, which has a temperature dependence. The
temperature-corrected snow height ($sh$) is calculated from
$$sh = sh_{initial} - sh_{raw} \times \sqrt{\frac{T_2 + 273.15}{273.15}} \times 100, \tag{i}$$

where $sh_{initial}$ (= 300 cm) is the initially installed sensor height from the surface and $T_2$ is air
temperature.

Table 1. Meteorological observation parameters and sensor specifications.

| observation parameter | abbreviation | unit | sensor | observaion range | accuracy |
|---|---|---|---|---|---|
| wind speed | $U_n$[a] | m s$^{-1}$ | Young, 05103 | 0 to 100 [m s$^{-1}$] | 1.0 m s$^{-1}$ [c] |
| wind direction | $WD_n$[a] | degree | Young, 05103 | 360° mechanical, 355° electrical (5° open) | 1.1 m s$^{-1}$ at 10° displacement[c] |
| air temperature | $T_n$[a] | °C | Vaisala, HMP155[b] | −80 to +60 [°C] | ±0.17 °C |
| relative humidity | $RH_n$[a] | % | Vaisala, HMP155[b] | 0 to 100% | ±1% (0 to 90%) ±1.7% (90 to 100%) |
| atmospheric pressure | $P_a$ | hPa | Vaisala, PTB210 | 500 to 1100 [hPa] | ±0.30 hPa at 20 °C |
| downward and upward shortwave radiation | $SW_d$, $SW_u$ | W m$^{-2}$ | Kipp & Zonen, CNR4 | 0.3 to 2.8 [μm] | 5 to 20 μV W$^{-1}$ m$^{-2}$ |
| downward and upward longwave radiation | $LW_d$, $LW_u$ | W m$^{-2}$ | Kipp & Zonen, CNR4 | 4.5 to 42 [μm] | 5 to 20 μV W$^{-1}$ m$^{-2}$ |
| downward and upward near-infrared radiation | $NIR_d$, $NIR_u$ | W m$^{-2}$ | Kipp & Zonen, CMP6 with a RG715 cut-off filter | 0.715 to 2.8 [μm] | 5 to 20 μV W$^{-1}$ m$^{-2}$ |
| snow height | $sh$ | cm | Campbell, SR50 | 0.5 to 10 [m] | 1 cm or 0.4% |
| snow temperature | $st_n$[a] | °C | Climatec, C-PTWP-10 | −40 to +60 [°C] | ±0.15°C |
| tilts of the main mast | $Tilt_X$, $Tilt_Y$ | degree | TURCK, B2N85H-Q20L60- | −85° to +85° | ±0.5° |

a: "n" suffix is appended to distinguish the observation height or depth.
b: protected from direct solar irradiance by a naturally-aspirated 14-plate Gill radiation shield
c: threshold sensitivity



## 3.2. Data processing

We describe the calculations for some variables used in the QC process in this section. Table 2
shows the key constants, variables, and abbreviations used in this study.
Because the vertical radiant flux against the inclined surface needed to accurately calculate the
surface albedo and surface energy balance is affected by the sloping surface at the SIGMA-B site, we
calculated the slope-corrected downward shortwave radiation ($SW_{\text{d\_slope}}$) from the corresponding
observations using the correction method in Jonsell et al. (2003) and Hock and Holmgren (2005). The
$SW_{\text{d\_slope}}$ is calculated by
$$SW_{\text{d\_slope}} = I_s + I_d, \qquad \text{(ii)}$$
where $I_s$ and $I_d$ are the direct and diffuse shortwave radiation for a slope, respectively:
$$I_s = SW_d \times d, \qquad \text{(iii)}$$
$$I_d = SW_d \times (1 - d) \times \frac{\cos \theta_{slope}}{\cos \theta}, \qquad \text{(iv)}$$
where $d$ is the ratio of total diffuse radiation to global radiation and $\theta$ and $\theta_{slope}$ [radian] are the solar
zenith angle and the solar zenith angle for a slope, respectively. The ratio $d$ is obtained from
atmospheric transmittance $t_r$ by
$$d = \begin{cases} 0.15 & \text{for } 0.8 \le t_r, \\ 0.929 + 1.134t_r - 5.111{t_r}^2 + 3.106{t_r}^3 & \text{for } 0.15 < t_r < 0.8, \\ 1.0 & \text{for } t_r \le 0.15, \end{cases} \qquad \text{(v)}$$
where
$$t_r = \frac{SW_d}{SW_{\text{TOA}}}, \qquad \text{(vi)}$$
where $SW_{\text{TOA}}$ is the downward shortwave radiation at the top of the atmosphere, calculated by
$$SW_{\text{TOA}} = I_0 \left(\frac{r_m}{r}\right)^2 \cos \theta, \qquad \text{(vii)}$$
where $I_0$ (= 1361 W m$^{-2}$) is the solar constant (Rottman, 2006; Fröhlich, 2012), $r$ is the distance
between the Sun and the Earth (assuming an elliptical orbit with an eccentricity of 0.01637), and $r_m$ is
its annual mean (= $1.496 \times 10^8$ km).
The solar zenith angle for a slope in Eq. (iv) is calculated by
$$\cos \theta_{\text{slope}} = \cos \beta \cos \theta + \sin \beta \sin \theta \cos(\varphi - \varphi_{\text{slope}}), \qquad \text{(viii)}$$
where $\beta$ is the slope angle from a horizontal plane, and $\varphi$ and $\varphi_{\text{slope}}$ are the solar azimuth and the solar
azimuth for the slope direction, respectively. Solar zenith and azimuth angles are calculated from the
geographic position of the observation site and the date and time.
Shortwave and near-infrared albedos ($\alpha_{\text{sw}}$ and $\alpha_{\text{nir}}$, respectively) are calculated as the ratio of



upward and downward radiant fluxes, as shown for $\alpha_{sw}$ by
$$\alpha_{sw} = \frac{SW_u}{SW_d},$$    (ix)
where $SW_u$ is the upward shortwave radiant flux and $SW_d$ is the downward shortwave radiant flux. The
daily integrated shortwave albedo ($\alpha_{sw,i}$) is calculated as the ratio of cumulative upward and
downward radiant fluxes for the past 24 h:
$$\alpha_{sw,i} = \sum_{24h} SW_u / \sum_{24h} SW_d.$$    (x)
The near-infrared albedo ($\alpha_{nir}$) and daily integrated near-infrared albedo ($\alpha_{nir,i}$) are calculated in the
same way. The near-infrared fraction is the ratio of the downward near-infrared radiant flux ($NIR_d$) to
$SW_d$.

Table 2. Key constants, variables, and their symbols used in this paper.



| symbol | name | value | unit |
|---|---|---|---|
| | constant | | |
| $f_{nir}$ | a fraction of near-infrared radiant flux in the shortwave radiant flux at the top of the atmosphere | 0.5151 | no dimension |
| $I_0$ | solar constant | 1361 | W m$^{-2}$ |
| n | cloud cover coefficient | 0.5 | no dimension |
| $r_m$ | annual mean distance between the Sun and the Earth | $1.496 \times 10^8$ | km |
| $sh_{initial}$ | initial height of the snow height sensor | 300 | cm |
| $\kappa$ | constant depending on cloud type | 0.26 | no dimension |
| $\varepsilon$ | snow/ice surface emissivity | 0.98 | no dimension |
| $\sigma$ | Stefan-Boltzmann constant | $5.67 \times 10^8$ | W m$^{-2}$ K$^{-4}$ |
| | variable | | |
| d | diffuse fraction in global radiation | | no dimension |
| $I_d$ | diffuse solar radiation | | W m$^{-2}$ |
| $I_s$ | direct solar radiation | | W m$^{-2}$ |
| $LW_d$ | downward longwave radiation | | W m$^{-2}$ |
| $LW_{std}$ | standard atmospheric longwave radiation | | W m$^{-2}$ |
| $LW_u$ | upward longwave radiation | | W m$^{-2}$ |
| $NIR_d$ | downward near-infrared radiation | | W m$^{-2}$ |
| $NIR_u$ | upward near-infrared radiation | | W m$^{-2}$ |
| $P_a$ | atmospheric pressure | | hPa |
| r | distance between the Sun and the Earth | | m |
| $RH_{1,2}$[a] | relative humidity | | % |
| sh | snow height | | cm |
| $sh_{raw}$ | raw data of snow height | | m |
| solz | solar zenith angle | | degree |
| $solz_{slope}$ | solar zenith angle for a slope | | degree |
| $st_{1-6}$[b] | snow temperature | | °C |
| $st\_depth_{1-6}$[b] | snow temperature sensor depth | | m |
| $SW_d$ | downward shortwave radiation | | W m$^{-2}$ |
| $SW_{d,slope}$ | downward shortwave radiation for a slope | | W m$^{-2}$ |
| $SW_{TOA}$ | downward shortwave radiation at the top of the atmosphere | | W m$^{-2}$ |
| $SW_u$ | upward shortwave radiation | | W m$^{-2}$ |
| $T_r$ | transmissivity of the atmosphere for shortwave radiation | | no dimension |
| $T_{1,2}$[a] | air temperature | | °C |
| $WD_{1,2}$[a] | wind direction | | degree |
| $U_{1,2}$[a] | wind speed | | m s$^{-1}$ |
| $\alpha_{sw}$ | surface albedo | | no dimension |
| $\alpha_{sw,i}$ | daily integrated surface albedo | | no dimension |
| $\alpha_{nir}$ | surface near-infrared albedo | | no dimension |
| $\alpha_{nir,i}$ | daily integrated surface near-infrared albedo | | no dimension |
| $\beta$ | slope angle | | radian |
| $\varepsilon_0$ | clear-sky atmospheric emissivity | | no dimension |
| $\varepsilon^*$ | atmospheric emissivity | | no dimension |
| $\theta$ | solar zenith angle | | radian |
| $\theta_{slope}$ | solar zenith angle for a slope | | radian |
| $\phi$ | solar azimuth angle | | radian |
| $\phi_{slope}$ | solar azimuth angle of a slope | | radian |

[a] 1: observed at lower height, 2: observed at upper height (only at the SIGMA-A site)

[b] 1-6: observing depth


## 4. Quality control


The datasets of observations at sites SIGMA-A and SIGMA-B are classified into four QC levels
numbered 1.0 to 1.3. A Level 1.0 dataset, which is not archived in any repository, is a raw dataset
without data processing. A Level 1.1 dataset is a raw dataset with flags added to indicate missing data
for periods when the data logger was inoperative. A Level 1.2 dataset has undergone an initial control,
which uses a simple masking algorithm to eliminate anomalous values that violate physical laws or
are impossible in the observed environment. The initial control improves the accuracy of the statistical
processing that follows and reduces the possibility of excluding true values. A Level 1.3 dataset has
undergone a secondary control, in which statistical methods are used on Level 1.2 data to identify and
mask outlier values. It has also undergone a final manual masking procedure, in which a researcher
visually checks the dataset and masks outliers based on subjective criteria.
The initial control method is described in Sect. 4.1 and the secondary control method is described
in Sect. 4.2. In these sections, the parameter suffixes related to the differences in observation height
(1 and 2) and sites (A and B) are omitted except when needed for clarity. Erroneous records are flagged
with one of the following numerical expressions to signify the reason they have been flagged:
−9999: a missing or erroneous data record attributed to a mechanical malfunction or a local
phenomenon such as sensor icing, riming, or burial in snow.
−9998: an erroneous radiation record when the radiant sensor was covered with snow or frost.
−9997: a record of snow temperature sensor depth when the sensor was suspected to be located above,
not below the snow surface.
−8888: a record flagged during the manual masking procedure.

### 4.1. Initial QC for Level 1.2 datasets


The objectives of the initial control are to eliminate erroneous records due to mechanical
malfunctions or local phenomena and pre-treat Level 1.1 datasets for the secondary control. The initial
control consists of a range test (e.g., Fiebrich et al., 2010; Estévez et al., 2011) and a manual mask
procedure. The range test sets variation ranges for each observed parameter in northwest Greenland
on the basis of simple statistics (maximum, minimum, and mean values) derived from records in the
Level 1.1 dataset during a period with no obvious erroneous data. Records outside this statistical range
are flagged with a "−9999" code. Table 3 lists the parameters subjected to this test and their assigned
ranges. The manual masking procedure identified specific erroneous values that resulted from an
electrical malfunction and flagged them with a "−8888" code. The following subsections offer detailed
and additional explanations of the initial control.



### 4.1.1. Wind speed and wind direction

The ranges for wind speed ($U_\mathrm{n}$) and wind direction ($WD_\mathrm{n}$) were set at

$$0 < U_\mathrm{n} < U_\mathrm{max} + 15.0, \tag{1.1.1}$$

$$0 < WD_\mathrm{n} \leq 360. \tag{1.1.2}$$

$U_\mathrm{max}$ is the maximum value between the beginning of observation and 31 August 2020, and +15.0 m s$^{-1}$ was taken as the range margin for the upper limit of $U_\mathrm{n}$. No data points for $U_\mathrm{n}$ were flagged by this initial control; however, the secondary control added a further condition that flagged erroneous values. When $U_\mathrm{n}$ was zero (no wind), $WD_\mathrm{n}$ was flagged as erroneous:

$$U_\mathrm{n} = 0 \ \text{ and } \ WD_\mathrm{n} > 0 \ \rightarrow WD_\mathrm{n} \text{ flagged } -9999. \tag{1.1.3}$$

When $WD_\mathrm{n}$ had a negative value, it was modified to zero:

$$WD_\mathrm{n} \leq 0 \ \rightarrow \ WD_\mathrm{n} = 0. \tag{1.1.4}$$

### 4.1.2. Air temperature and relative humidity

The ranges for air temperature ($T_\mathrm{n}$) and relative humidity ($RH_\mathrm{n}$) were set at

$$T_\mathrm{n\_min} - 10.0 < T_\mathrm{n} < T_\mathrm{n\_max} + 10.0, \tag{1.2.1}$$

$$0 \leq RH_\mathrm{n} \leq 100. \tag{1.2.2}$$

$T_\mathrm{n\_max}$ and $T_\mathrm{n\_min}$ were determined from the observation period ending 31 August 2020. The range margin for $T_\mathrm{n}$ was set as ±10.0 °C. Discrepancies arising from the dual sensors at SIGMA-A were addressed in the secondary control (see Sect. 4.2.2).

### 4.1.3. Shortwave and near-infrared radiation

The main objective of the initial control for shortwave radiation was to mask erroneous records attributed to electrical noise. The range test is based on the assumption that $SW_\mathrm{d}$ cannot exceed the maximum of $SW_\mathrm{TOA}$ ($SW_\mathrm{TOA\_max}$) during the observation period (761.6 W m$^{-2}$ at SIGMA-A and 772.2 W m$^{-2}$ at SIGMA-B), and albedos $\alpha_\mathrm{sw}$ and $\alpha_\mathrm{nir}$ cannot be lower than 0.95 and 0.90, respectively, as determined from the radiative transfer model calculation (Aoki et al., 2003). Moreover, the fraction of the near-infrared spectral domain at the top of the atmosphere ($f_\mathrm{nir}$) is assumed to be equal to 0.5151 based on the extraterrestrial spectral solar radiation (Wehrli, 1985). Based on those assumptions, upward and downward radiation fluxes were flagged as erroneous (–9999) according to the following criteria:

$$SW_\mathrm{d} < SW_\mathrm{TOA\_max}, \tag{1.3.1}$$

$$NIR_\mathrm{d} < f_\mathrm{nir}\, SW_\mathrm{TOA\_max}, \tag{1.3.2}$$

$$SW_\mathrm{u} < 0.95\, SW_\mathrm{TOA\_max}, \tag{1.3.3}$$

$$NIR_\mathrm{u} < 0.90\, f_\mathrm{nir}\, SW_\mathrm{TOA\_max}. \tag{1.3.4}$$

The following procedures were also applied to mask erroneous records due to electrical noise.



These parameters were flagged as erroneous ($-9999$) when
$$(SW_d, SW_u, NIR_d, NIR_u) < 0 \text{ and } solz < 90.0, \qquad (1.3.4)$$
and were changed to zero when
$$(SW_d, SW_u, NIR_d, NIR_u) < 0 \text{ and } solz \geq 90.0. \qquad (1.3.5)$$

### 4.1.4. Longwave radiation

The ranges for $LW_d$ and $LW_u$ were set as follows:
$$0 < LW_d \ (LW_u) < LW_{d\_max} \ (LW_{u\_max}), \qquad (1.4.1)$$
where
$$LW_{d\_max} = \varepsilon_{max} \sigma \, T_{2A\_max} \ (T_{1B\_max}), \qquad (1.4.2)$$
$$LW_{u\_max} = \varepsilon \sigma \, T_{s\_max}. \qquad (1.4.3)$$
Maximum values were determined under the following assumptions: (1) $T_{2A}$ and $T_{1B}$ cannot be larger
than $T_{2A\_max}$ and $T_{1B\_max}$, respectively, (2) atmospheric emissivity is set to unity ($\varepsilon_{max}$), and (3) the value
of $LW_{u\_max}$ is determined by assuming that the surface temperature cannot exceed $T_{s\_max}$ (= 10 °C),
which includes errors due to longwave emissions from the poles of the AWS system and similar
sources, and that the emissivity of the snow/ice surface ($\varepsilon$) is 0.98 (Armstrong and Brun, 2008).
Both upward and downward longwave fluxes were considered erroneous when the sensor appeared
to be covered with snow or frost:
$$|LW_d - LW_u| \leq 1.0 \ \rightarrow \ LW_d \text{ and } LW_u \text{ flagged } -9998. \qquad (1.4.4)$$

### 4.1.5. Snow height

The range test for snow height ($sh$) was imposed separately for each period between maintenances
to the SIGMA-A site, when the mainmast extension was adjusted to prevent the sensors from being
buried in snow. (A single range test sufficed for SIGMA-B.) For each test, the range was set so that $sh$
varied from the median by ±100 cm or ±150 cm, a margin that was determined depending on the
variation of the data records in each period. The objective was to mask the most obvious outliers. In
addition, corrections were made to the $sh$ records after each of three maintenance visits to the AWS at
SIGMA-A.

### 4.1.6. Atmospheric pressure

The range test for atmospheric pressure ($P_a$) was conducted according to
$$P_{a\_ave} - 100.0 < P_a < P_{a\_ave} + 100.0, \qquad (1.6.1)$$
where $P_{a\_ave}$ is the average atmospheric pressure for the observation period at each AWS site (Table
3). The additional margin that defined the range was ±100 hPa.





**4.1.7. Snow temperature**
The range test for snow temperature ($st_n$) was conducted according to
$T_{1\_min} < st_n < 0.2,$                                                      (1.7.1)
where $T_{1\_min}$ is the minimum air temperature for the site and the upper threshold, 0.2 ℃, incorporates
the sensor's absolute error of 0.15 ℃ and the requirement that the snow temperature cannot be positive.

Table 3. Threshold values used in the range tests, determined from the entire observation period up to
31 August 2020.

| meteorological parameter | unit | threshold value | | | |
|---|---|---|---|---|---|
| | | SIGMA-A | | SIGMA-B | |
| | | parameter name | value | parameter name | value |
| wind speed | m s$^{-1}$ | $U_{1A\_max}$ | 23.9 | $U_{1B\_max}$ | 21.9 |
| | | $U_{2A\_max}$ | 25.5 | − | − |
| air temperature | ℃ | $T_{1A\_max}$ | 7.2 | $T_{1B\_max}$ | 10.7 |
| | | $T_{2A\_max}$ | 7.2 | − | − |
| | | $T_{1A\_min}$ | −49.9 | $T_{1B\_min}$ | −40.5 |
| | | $T_{2A\_min}$ | −49.9 | − | − |
| longwave radiation | W m$^{-2}$ | $LW_{dA\_max}$ | 418.8 | $LW_{dB\_max}$ | 440.1 |
| | | $LW_{uA\_max}$ | 357.2 | $LW_{uB\_max}$ | 357.2 |
| atmospheric pressure | hPa | $P_{a\_aveA}$ | 833.1 | $P_{a\_aveB}$ | 894.2 |


**4.2. Secondary QC for Level 1.3 datasets**
The secondary control applies another range test, an anomaly test, and a manual mask procedure.
The range test, applied only to the shortwave radiation and albedo data, sets a more precise variation
range than the initial control and masks erroneous data records. The anomaly test sets a median and
standard deviation (SD), which govern statistical tests used to determine the possible range of normal
values in the Level 1.2 dataset and identify and mask outliers. The manual mask procedure identifies
and masks any remaining erroneous records. The effects of the initial and secondary controls are
illustrated in Fig. 3 and described in detail below. .

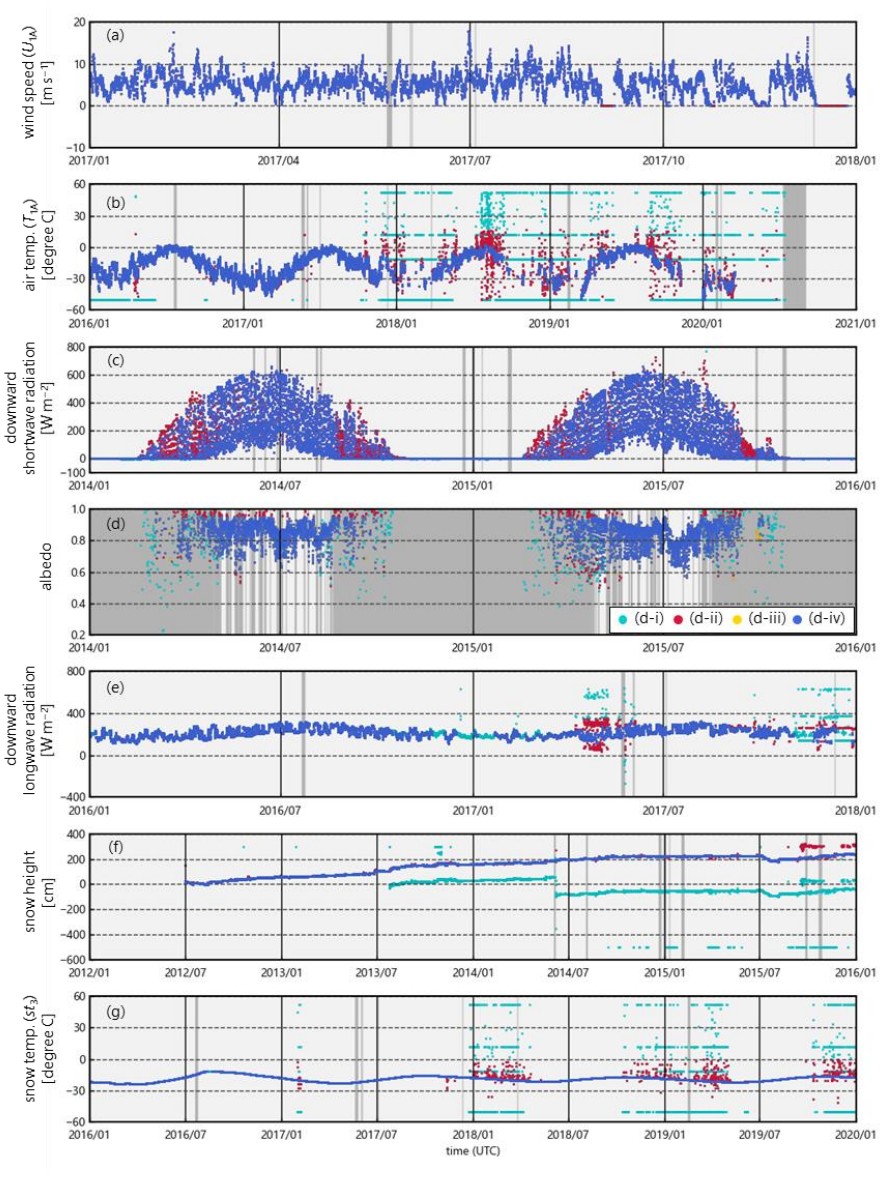


Figure 3. Examples of the initial and secondary controls for the SIGMA-A site: (a) wind speed ($U_{1A}$),
(b) air temperature ($T_{1A}$), (c) downward shortwave radiation, (d) surface albedo, (e) downward
longwave radiation, (f) snow height, and (g) snow temperature ($st_3$). In all panels except (d), the dark
gray areas represent time periods in which data records in the Level 1.0 dataset were masked to
produce the Level 1.1 dataset, light blue dots denote records masked by the initial control, red dots



denote records masked by the secondary control, and dark blue dots are the Level 1.3 data records. In
panel (d), the gray shaded area represents the masked ($-9999$) data records that cannot be calculated
due to the absence of, masked $SW_d$, or for other reasons. The light blue, red and yellow dots represent
data points masked by three QC operations during the secondary control; see Sect. 4.2.4 for
explanation.

### 4.2.1. Wind speed and wind direction


When $U_n$ was zero for more than 6 continuous hours, $U_n$ and $WD_n$ were both flagged as erroneous
($-9999$) under the assumption that the wind sensor was blocked by snow and ice. Although the initial
control eliminated no $U_n$ records, this step masked many values in the winter (Fig. 3a).

### 4.2.2. Air temperature and relative humidity


Anomaly tests for air temperature and relative humidity were only applied to the lower-level
sensor records for SIGMA-A (i.e., $T_{1A}$ and $RH_{1A}$). The anomaly test compared the difference
($\Delta T$ and $\Delta RH$) between readings of the upper and lower sensors (i.e., $|T_{1A} - T_{2A}|$ and $|RH_{1A} -$
$RH_{2A}|$) to the respective medians and SDs of those parameters:
$\Delta T <$ median_$\Delta T$ + SD_$\Delta T \times 3$    for before 1 September 2017,    (2.2.1)
$\Delta T <$ median_$\Delta T$ + SD_$\Delta T$      for after 1 September 2017,    (2.2.2)
$\Delta RH <$ median_$\Delta RH$ + SD_$\Delta RH \times 3$.    (2.2.3)
The medians were calculated from the data before 1 September 2017, because the data after that date
appeared to include many erroneous $T_{1A}$ records due to deterioration of the data logger or sensor. For
these later records, the SD criterion was adjusted to more stringently detect outliers in the records of
$T_{1A}$ and $RH_{1A}$, which were flagged as erroneous ($-9999$). The effectiveness of this adjustment is clear
in Fig. 3b.

### 4.2.3. Shortwave and near-infrared radiation


The anomaly test for shortwave and near-infrared radiation was intended to mask the noise
resulting from a weak electric pulse at large solar zenith angles. The median and SD values were
calculated from only the records ($SW_d$, $SW_u$, $NIR_d$, and $NIR_u$) at $solz > 90.0°$ to distinguish this noise
source according to the following, using $SW_d$ as an example:
$SW_d <$ median_$SW_d$ + SD_$SW_d \times 3$.    (2.3.1)
Records identified as noise were modified to zero.
The downward radiation components were sometimes overestimated as a result of icing or riming
over the glass dome of the pyranometer. To mask these erroneous values, we applied range tests based
on $SW_{TOA}$ and a threshold value of atmospheric transmittance $T_r$ (0.881 for SIGMA-A and 0.872 for



SIGMA-B) calculated by a radiative transfer model (Aoki et al., 1999, 2003):
$$SW_d < T_r\,SW_{TOA},\qquad\qquad(2.3.2)$$
$$NIR_d < T_r\,f_{nir}\,SW_{TOA}.\qquad\qquad(2.3.3)$$
Values of $SW_d$ and $NIR_d$ that were outside this range were flagged as erroneous ($-9999$).
To recognize other instances when the radiation sensor was covered with snow or frost, $SW_d$ and
$NIR_d$ records corresponding to the following case were flagged as erroneous ($-9998$):
$$SW_d\,(NIR_d) < SW_u(NIR_u).\qquad\qquad(2.3.4)$$
Figure 3c shows that the initial control eliminated a few erroneous $SW_d$ data recorded in August 2015,
whereas the secondary control masked many records, especially in February–May, that were affected
by riming or frost.
**4.2.4. Shortwave and near-infrared albedo**
We calculated albedos $\alpha_{sw}$ and $\alpha_{nir}$, and the statistical values used in all QC procedures for
those albedos, from the $SW_d$ and $NIR_d$ datasets that had first undergone secondary control. This
calculation was done in four separate steps, shown by the color of dots in Fig. 3d.
(1) Flagging for low pyranometer sensitivity
At solar zenith angles near 90.0°, $SW_d$ and $NIR_d$ may not be an accurate measurement because of
the low sensitivity of the pyranometer. We therefore masked $\alpha_{sw}$ and $\alpha_{nir}$ values at $solz > 85.0°$ or
when the $SW_d\,(NIR_d)$ value was below the median $SW_d\,(NIR_d)$ value for $solz > 85.0°$. Records masked
in this step are shown in Fig. 3d as light blue dots (d-i).
(2) Range test for cold and warm periods
The range test used the upper and lower thresholds for $\alpha_{sw}$ and $\alpha_{nir}$, as determined by the
radiative transfer calculation of Aoki et al. (2003, 2011) plus a small error margin. Those thresholds
correspond to the assumed surface conditions during two parts of the year. For the cold period of
October–April, we used the following thresholds for different snow or ice conditions:
$\quad0.6 < \alpha_{sw} < 0.95\qquad$ for dry snow at SIGMA-A,$\qquad\qquad$(2.4.1)
$\quad0.5 < \alpha_{nir} < 0.90\qquad$ for dry snow at SIGMA-A,$\qquad\qquad$(2.4.2)
$\quad0.4 < \alpha_{sw} < 0.95\qquad$ for dry or wet snow at SIGMA-B.$\qquad\qquad$(2.4.3)
For the warm period of May–September we used the following thresholds:
$\quad0.4 < \alpha_{sw} < 0.95\qquad$ for wet snow at SIGMA-A,$\qquad\qquad$(2.4.4)
$\quad0.3 < \alpha_{nir} < 0.90\qquad$ for wet snow at SIGMA-A,$\qquad\qquad$(2.4.5)
$\quad0.1 < \alpha_{sw} < 0.95\qquad$ for wet snow or dark ice at SIGMA-B.$\qquad\qquad$(2.4.6)
Records with albedo values beyond these theoretical thresholds were masked.
(3) Anomaly test in low atmospheric transmittance condition
The range test was augmented by an anomaly test to identify underestimates of $\alpha_{sw}$ and $\alpha_{nir}$
when $SW_d\,(NIR_d)$ was low and atmospheric transmittance ($t_r$) was small, typically at large solar zenith



angles. Whereas the first QC step in this phase used a criterion of $solz > 85.0°$, we relaxed it to $solz >$
$80.0°$ and masked $\alpha_{sw}$ ($\alpha_{nir}$) values that were unnaturally low owing to low $t_r$ and $SW_d$ ($NIR_d$). Data
records that were masked in either the range or anomaly tests are shown in Fig. 3d as red dots (d-ii).
(4) Final steps
In cases where $LW_d$ was flagged as "−9998" during the initial control (see Sect. 4.1.4), $\alpha_{sw}$ and
$\alpha_{nir}$ were flagged as "−9999" under the assumption that the radiation sensors were covered with snow
or frost. The final step was a manual mask procedure. Data records that were masked in this phase are
shown in Fig. 3d as orange dots (d-iii), and the final Level 1.3 dataset is displayed as blue dots (d-iv).
**4.2.5. Longwave radiation**

The anomaly test for $LW_d$ and $LW_u$ was conducted only for the SIGMA-A dataset using a standard
longwave radiant flux ($LW_{std}$), a measure of the amount of longwave radiation from the near-surface
atmosphere that was calculated from the air temperature measurement by Brock and Arnold (2000)
$$LW_{std} = \varepsilon^* \sigma (T_{2A} + 273.15)^4, \text{(xi)}$$
$$\varepsilon^* = (1 + \kappa n)\varepsilon_0, \text{(xii)}$$
$$\varepsilon_0 = 8.733 \times 10^{-3} \times (T_{2A} + 273.15)^{0.788}, \text{(xiii)}$$
where $\varepsilon^*$ is the atmospheric emissivity, $\sigma$ ($= 5.670 \times 10^{-8}$) is the Stefan–Boltzmann constant, $\kappa$
($= 0.26$) is a constant depending on cloud type (Braithwaite and Olsen, 1990), $n$ is the cloud cover
amount ($n$: [0, 1] and set at 0.5 because it could not be determined), and $\varepsilon_0$ is the clear-sky emissivity.
We assumed that $LW_{std}$ was a close approximation of the true longwave radiant fluxes and used the
absolute difference between $LW_{std}$ and $LW_d$ or $LW_u$ (i.e., $\Delta LW_d$ or $\Delta LW_u$) and its median and SD as
the basis of the anomaly test.
Because parts of the $LW_d$ dataset contained many erroneous records attributed to degradation of
the data logger (see Fig. 3e), we reduced the SD range limit by half for two time periods, 7 April to 7
June 2017 and after 1 September 2017. The resulting criteria were
$$\Delta LW_d < \text{median\_}\Delta LW_d + \text{SD\_}\Delta LW_d \times 2 \quad \text{for all periods, except} \text{(2.5.1)}$$
$$\Delta LW_d < \text{median\_}\Delta LW_d + \text{SD\_}\Delta LW_d \quad\quad \text{for two subperiods,} \text{(2.5.2)}$$
$$\Delta LW_u < \text{median\_}\Delta LW_u + \text{SD\_}\Delta LW_u \times 2 \quad \text{for all periods.} \text{(2.5.3)}$$
Records that were outliers under these criteria were flagged as erroneous (−9999). Figure 3e shows
that the initial control (see Sect. 4.1.4) improved this anomaly test's efficacy, and the secondary control
yielded a clean $LW_d$ time series.
**4.2.6. Snow height**

The anomaly test for snow height masked data that displayed unrealistic fluctuations.
Differences ($\Delta sh$) were determined with respect to mean and SD values from the preceding 72 h values
during period 1, before 1 September 2017 ($sh_{mean1}$) and period 2, after 1 September 2017 ($sh_{mean2}$). The





difference between the two periods is that means were not calculated when the 72 h period included
more than 48 flagged records in period 1 and more than 60 flagged records in period 2. The $\Delta sh$
values were compared to the median plus SD of $\Delta sh$ for that period. In addition, because snow height
increased steadily in period 2, we derived the regression equation for this increase and identified
outliers with respect to the SD of the regression, i.e. $\Delta sh_{\mathrm{reg}}$. The resulting criteria were

$\Delta sh_{\mathrm{mean1}} < \mathrm{median}_1\_\Delta sh + \mathrm{SD}_1 \Delta sh,$                         (2.6.1)

$\Delta sh_{\mathrm{reg}} < \mathrm{SD}_{\mathrm{reg}}\_sh$                              for after 1 September 2017,   (2.6.2)

$\Delta sh_{\mathrm{mean2}} < \mathrm{median}_2\_\Delta sh + \mathrm{SD}_2 \Delta sh \times 3$    for after 1 September 2017.   (2.6.3)

Records of $sh$ that varied beyond these threshold values were flagged as erroneous (−9999).

A manual mask procedure was added as a final step. The result of QC procedure is shown in Fig.

3f. The initial control, which corrected gaps resulting from the AWS maintenance (see Sect. 4.1.5),
yielded the smoothed data record that enabled the application of the anomaly test.

### 406    4.2.7. Snow temperature

In the first step, data records were masked when the snow temperature sensor was suspected to be

located above the snow surface:

$st\_depth_{\mathrm{n}} < -1.0 \; \rightarrow \; st_{\mathrm{n}}$  flagged −9999.                          (2.7.1)

where $st\_depth_{\mathrm{n}}$ was calculated using snow height data and the initial setting depth of sensor "n" (see
Sect. 3). The threshold of $st\_depth_{\mathrm{n}}$ included a margin of 1.0 cm to reflect the accuracy of the snow
height sensor. The $st_{\mathrm{n}}$ was flagged as "−9997" if we could not judge whether the snow temperature
sensor was located below the snow surface.

The anomaly test for $st_{\mathrm{n}}$ consisted of two procedures. The first procedure relied on a temperature

gap between $st_4$ and data from each of the other five levels ($st_{\mathrm{not4}}$), because $st_4$ had very few erroneous
data:

$|st_4 - st_{\mathrm{not4}}| > \mathrm{median}\_st_4 + \mathrm{SD}\_st_{\mathrm{not4}} \times y.$                   (2.7.2)

where the multiplier $y$ is 1, 2, or 3 depending on the intensity of the anomaly variation, and determined
based on the test results in each case.

The second procedure used the difference between $st_{\mathrm{n}}$ and its mean value $st_{\mathrm{n\_mean}}$ from the

previous 72 h, calculated using the same method as $sh_{\mathrm{mean}}$ (see Sect. 4.2.6):

$\left|st_{\mathrm{n}} - st_{\mathrm{n\_mean}}\right| > \mathrm{median}\_st_{\mathrm{n\_mean}} + \mathrm{SD}\_st_{\mathrm{n\_mean}}.$              (2.7.3)

In both procedures, the median and SD terms were calculated from records for the full time period.
Records detected as outliers were flagged as "−9999". Figure 3g shows the results of all procedures,
using $st_3$ as an example.

### 426    4.2.8. Atmospheric pressure

The time series of $P_{\mathrm{a}}$ included only a few erroneous records. We masked outliers on the basis of



$|P_\mathrm{a} - P_\mathrm{a\_mean}| > 20.0,$                                                                      (2.8.1)
where $P_\mathrm{a\_mean}$ is the average for the past 3 h (excluding masked data records). We set the threshold at
20.0, a higher value than the SD, because using the SD could have masked valid records.
**5. Temporal variations of meteorological parameters**
This section shows the results of simple analyses of the Level 1.3 dataset.
**5.1. Air temperature and snow height**
Figure 4 shows the air temperature fluctuations and snow height (*sh)* variations at both sites. Mean
air temperatures (2013–2019) were −18.1 °C at the SIGMA-A site and −12.3 °C at the SIGMA-B site.
The annual maxima were recorded every July at both sites, except for August 2019 at the SIGMA-B
site. In contrast, the annual minima occurred in different months between December and March. The
maximum was slightly positive at the SIGMA-A site, and it was above freezing in all years at the
SIGMA-B site. Unusually high temperatures were recorded in mid-July 2015 (7.2 °C at SIGMA-A
and 10.7 °C at SIGMA-B). Air temperatures exceeding 5.0 °C at SIGMA-A and 10.0 °C at SIGMA-B
were common during that period.
Warm summers were observed at both sites in 2015, 2016, 2019, and 2020, as indicated by the
cumulative positive degree-day (PDD) records in Fig. 5. PDD generally increased after mid-June and
significantly ascended from late June to July. This tendency was especially strong in warmer years.
PDDs were an order of magnitude greater at SIGMA-B than at SIGMA-A. They increased gradually
until late August at SIGMA-B, whereas the increases at SIGMA-A were stepwise and stopped earlier,
in mid to late July.
Snow height steadily increased at the SIGMA-A site during the 8-year study period (Fig. 4), in
which *sh* rose approximately 1 m in the mass-balance years (September to August) of 2013/14,
2016/17, and 2017/18, and decreased slightly in the summers of 2011/12, 2014/15, and 2019/20.
Accumulations were notable in autumn and relatively small in winter. At the SIGMA-B site, in contrast,
increases and decreases in *sh* were observed during each mass-balance year. Decreases in *sh* during
summers were rare during the summers of 2012/13 and 2017/18 but common during the 2013/14,
2014/15, 2015/16, 2018/19, and 2019/20 summers, when decreases were greater than 1 m.

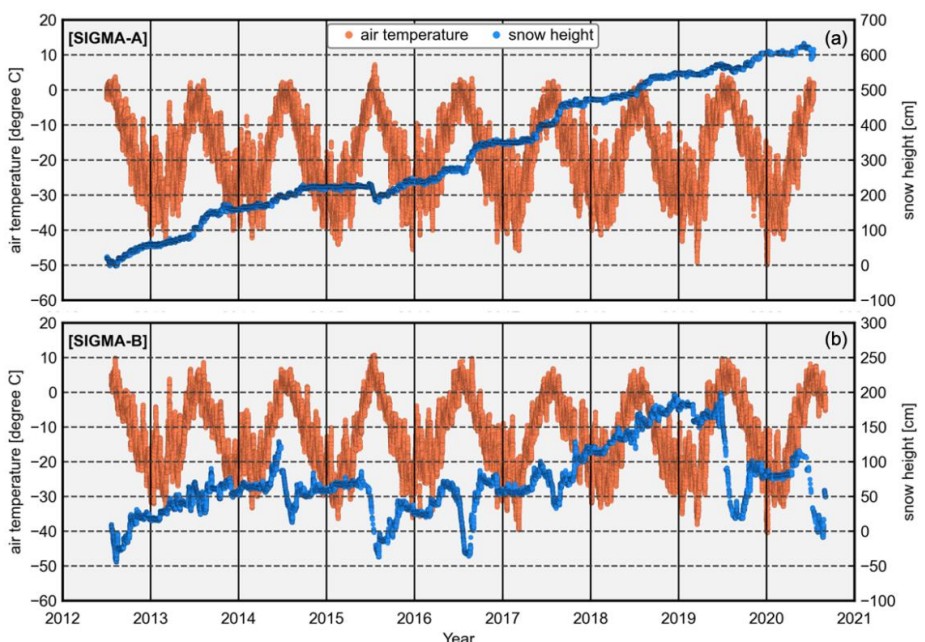


Figure 4. Time series of hourly air temperature and snow height at the (a) SIGMA-A (showing $T_2$ data)

and (b) SIGMA-B sites.

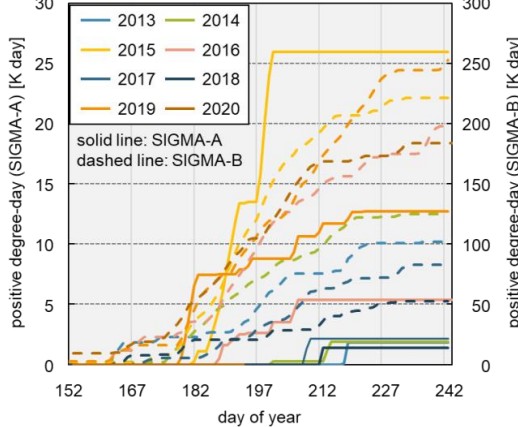


Figure 5. Cumulative positive degree-days at the SIGMA-A (solid lines) and SIGMA-B (dashed lines)

sites from 1 June to 31 August, 2013–2020.

## 5.2. Atmospheric pressure and seasonal variation of temperature lapse rate

The time series of atmospheric pressure ($P_a$) at the SIGMA-A and SIGMA-B sites show a clear seasonal variation, high in summer and low in winter (Fig. 6). The two data records had similar variation patterns that were strongly correlated ($r = 0.98$). The mean values for the whole observation period were 833.1 hPa at site SIGMA-A and 894.2 hPa at site SIGMA-B (Table 3). The difference in monthly mean $P_a$ between the sites was smaller in summer and larger in winter (Fig. 7a), and the amplitude of the annual cycle was greater at the SIGMA-A site.

The apparent lapse rate, indicated by the difference in monthly mean air temperatures between the elevations of the SIGMA-A and SIGMA-B sites, was approximately 8 K km$^{-1}$ in June and July and approximately 12 K km$^{-1}$ in November–February (Fig. 7b). Factors in summer that may contribute to this seasonal difference include a smaller difference in $P_a$ between the two sites and moister atmospheric conditions. The greater annual range of monthly air temperature at site SIGMA-A than at site SIGMA-B is likely also a winter effect. Winter is colder at SIGMA-A than at SIGMA-B because the SIGMA-A site is at a higher elevation and farther inland, where cooling by longwave emissions from the surface is greater and heat advection from the ocean is smaller. The temperature difference may lead in turn to a larger atmospheric pressure difference between the two sites in winter through its effect on atmospheric density. The combined summer and winter effects may be the reason that the apparent lapse rate is greater than the adiabatic reduction rate of the atmosphere (5 K for wet conditions and 10 K for dry conditions) (Fig. 7b).

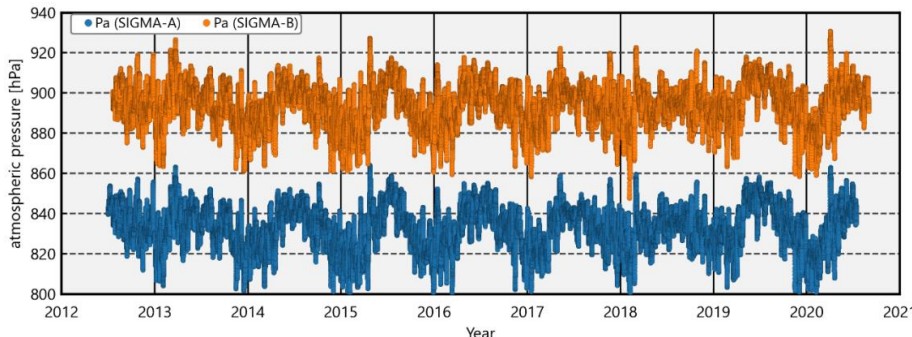

Figure 6. Time series of hourly atmospheric pressure ($P_a$) at the SIGMA-A and SIGMA-B sites.

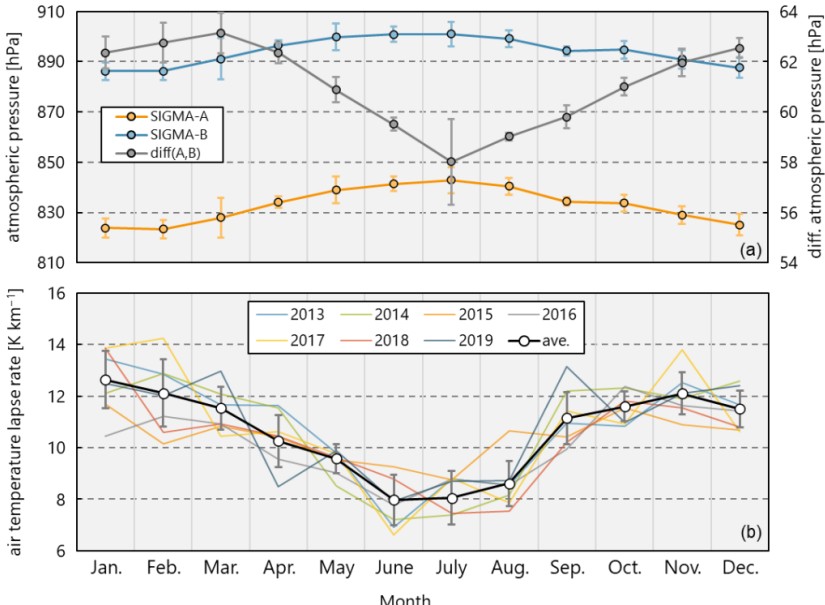


Figure 7. Time series of (a) ensemble averages of monthly mean atmospheric pressures during all
years at both sites and their difference and (b) monthly mean lapse rates of air temperature between
the SIGMA-A and SIGMA-B sites for each year (colored lines) and their ensemble average during all
years (open circles). Error bars indicate ±1 SD.

**5.3. Albedo**
Whereas shortwave albedo ($\alpha_{sw}$) was rarely lower than 0.7 at site SIGMA-A, near-infrared albedo
($\alpha_{nir}$) was below 0.6 in 2012, 2015, 2016, 2019, and 2020 (Fig. 8). Because $\alpha_{nir}$ depends on the
snow grain size (Wiscombe and Warren, 1980), this finding implies that snow metamorphism
progressed at the SIGMA-A site in those years (Hirose et al., 2021). A strong decrease in $\alpha_{sw}$ was
observed at the SIGMA-B site during those same summers, which corresponded to notable decreases
in snow height (Fig. 4b) and high PDDs (Fig. 5). The decreases in albedo may have accelerated
snowmelt and caused the decreases in snow height at SIGMA-B during the warm summers of those
years (see Sect. 5.1). It appears that the difference in albedo reduction between the SIGMA-A and
SIGMA-B sites in summer originated from the difference in air temperature between the sites.

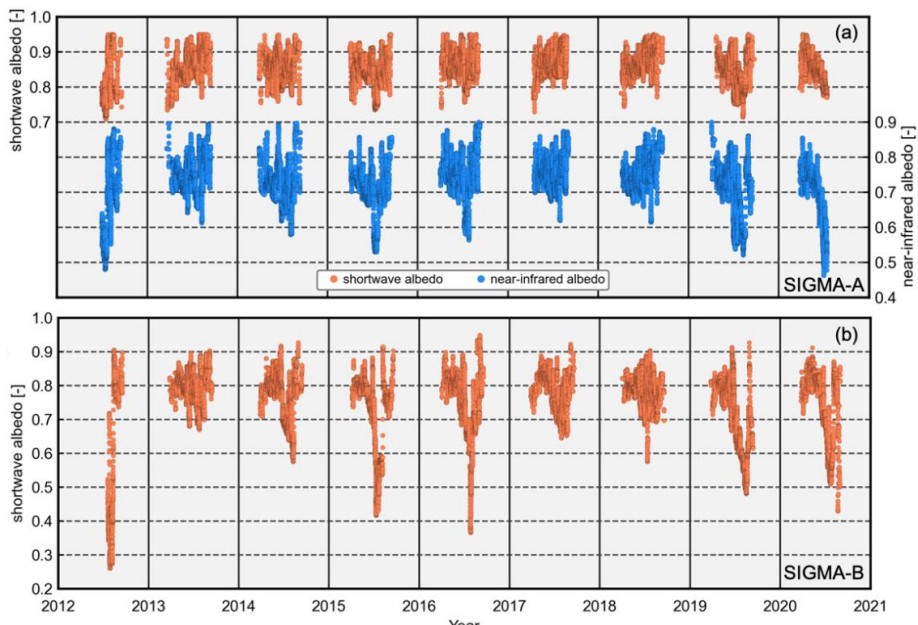


Figure 8. Time series of hourly shortwave and near-infrared albedos at the (a) SIGMA-A and (b)
SIGMA-B sites.

**5.4. Snow temperature**
Figure 9 shows the time series of snow temperatures ($st_1$–$st_6$) and snow sensor depths ($st\_depth_{1-}$
$_6$). The sensor depths were calculated from each sensor's initial depths (see Sect. 3.1) and the snow
height variations at the SIGMA-A site. Seasonal and short-term snow temperature fluctuations were
observed, which became smaller after the 2016/17 winter season, when snow accumulation was very
large (Fig. 4). We assumed that the sensors were buried more deeply at that time, resulting in smaller
fluctuations in snow temperature. The annual mean snow temperatures after 2016, a year in which
snow temperatures were relatively stable and less variable, were between −18.9 ± 0.5 °C ($st_4$) and
−19.5 ± 1.7 °C ($st_5$).
Sensors recorded relatively high snow temperatures when they were positioned at shallow depths
below the snow surface. However, in the summer of 2015, sensors $st_3$ and $st_4$ registered 0 °C even
though they were more than 1 m below the snow surface. Air temperatures above freezing, and a large
decrease in snow height were observed in this period (Figs. 4 and 5); thus, it is plausible that snowmelt
occurred from the surface to depths near 120 cm, where $st_3$ was located at that time.



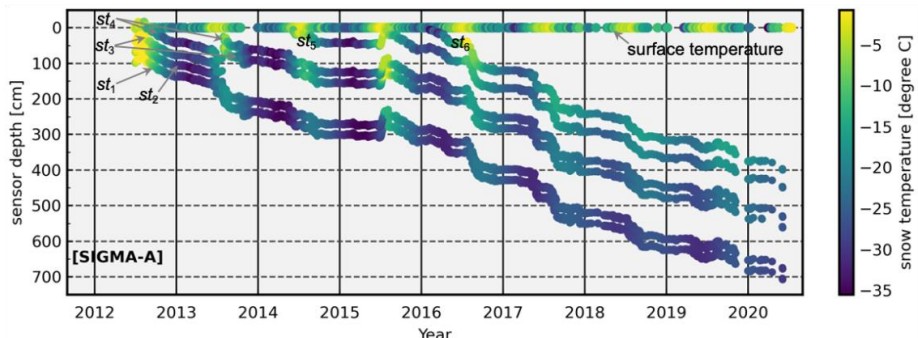

Figure 9. Time series of hourly snow temperatures ($st_1$–$st_6$), sensor depth, and surface temperature
(calculated from upward longwave radiation) at the SIGMA-A site.

### 5.5. Longwave radiation

The occurrence frequency of longwave radiation, taken to represent the atmospheric condition, is
often used as an indicator of climatological cloudiness (Stramler et al., 2011). Figure 10 shows the
histograms of occurrence frequency of downward ($LW_d$) and net longwave radiation ($LW_{net} = LW_d -$
$LW_u$) during July of all years at the SIGMA-A and SIGMA-B sites. The corresponding histograms for
the four seasons (autumn: SON, winter: DJF, spring: MAM, summer: JJA) are shown in Figs. S1 and
S2. The July $LW_d$ data from both sites had bimodal distributions, with a lower mode of 220–240 W
m$^{-2}$ at SIGMA-A and 240–260 W m$^{-2}$ at SIGMA-B, and a higher mode of 290–310 W m$^{-2}$ at SIGMA-
A and 310–330 W m$^{-2}$ at SIGMA-B. The histograms of July and seasonal $LW_{net}$ had similar but clearer
bimodal distributions, with modes at approximately 0 W m$^{-2}$ and −70 W m$^{-2}$ (Figs. 10c-d and S2).
$LW_{net}$ can be regarded as an indicator of cloudiness, which can significantly change the downward
longwave radiation and thus the surface temperature of the snow or ice. Both downward and net
longwave radiation increase under overcast conditions because of blackbody radiation from the cloud
cover that is absent in clear-sky conditions. Stramler et al. (2011) and Morrison et al. (2012) have
argued that surface net longwave radiative flux has two modes in occurrence frequency (at −40 W m$^{-2}$
and 0 W m$^{-2}$), which correspond to clear-sky and overcast (low-level mixed-phase clouds) conditions.
In overcast conditions, because the cloud base and the surface are in thermal equilibrium, the vertical
thermal gradient is small and the longwave radiation budget is balanced ($LW_{net} = 0$ W m$^{-2}$) at the
surface. The two modes of $LW_{net}$ (0 W m$^{-2}$ and −70 W m$^{-2}$) at the two AWS sites appear to correspond
to the modes proposed by these earlier studies.
The occurrence frequency of $LW_{net}$ in JJA appears to be more variable than those for the other
seasons at both sites (Fig. S2). In these months, the air temperature rises and sea ice extent decreases,
increasing the water vapor supply and advection from the surrounding sea to coastal Greenland (Kim
and Kim, 2017; Liang et al., 2022). In such atmospheric conditions, the cloud formation process is
susceptible to synoptic-scale disturbances. The histogram of $LW_{net}$ for July (Fig. 10) indicates clear-
sky ($LW_{net} \cong -70$ W m$^{-2}$) in 2015, 2019, and 2020 and overcast conditions ($LW_{net} \cong 0$ W m$^{-2}$) in 2014
and 2018. In contrast, annual occurrence frequencies for SON and MAM were less variable than those
for JJA. Overcast and clear-sky conditions dominated in SON and MAM, respectively. Our analysis
shows that cloudiness in JJA was more variable than in other seasons, a result that is also borne out by
satellite observations (Ryan et al., 2022).

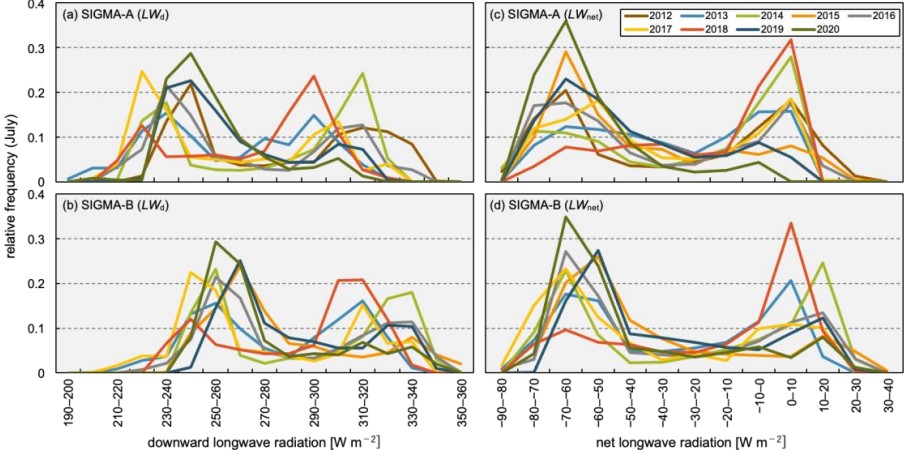


Figure 10. Histograms of the occurrence frequency of hourly downward longwave radiation ($LW_d$) and
net longwave radiation ($LW_{net}$) observed at the SIGMA-A and SIGMA-B sites in July of all years in
the study period. Each relative frequency represents the fraction of the total contained in each 10 W
m$^{-2}$ bin.
**6. Data availability**
The Level 1.1, 1.2, and 1.3 datasets from this study are archived and available from the Arctic Data
archive System (ADS) in the National Institute of Polar Research (Table 4), where they are stored in
text (CSV) file format. Detailed information on the data content is presented in the file
"data_format_*site-name_data-level*.csv" associated with each of these dataset files.

Table 4. Information for the archived datasets from the SIGMA-A and SIGMA-B sites.



| **SIGMA-A** | |
|---|---|
| Level 1.1 | |
| data name: | Quality-controlled datasets of Automatic Weather Station (AWS) at SIGMA-A site from 2012 to 2020: Level: 1.1 |
| file name: | SIGMA_AWS_SiteA_2012-2020_Lv1_1.csv |
| citation: | http://doi.org/10.17592/001.2022041301 |
| reference: | Nishimura et al. (2023a) |
| Level 1.2 | |
| data name: | Quality-controlled datasets of Automatic Weather Station (AWS) at SIGMA-A site from 2012 to 2020: Level: 1.2 |
| file name: | SIGMA_AWS_SiteA_2012-2020_Lv1_2.csv |
| citation: | http://doi.org/10.17592/001.2022041302 |
| reference: | Nishimura et al. (2023b) |
| Level 1.3 | |
| data name: | Quality-controlled datasets of Automatic Weather Station (AWS) at SIGMA-A site from 2012 to 2020: Level: 1.3 |
| file name: | SIGMA_AWS_SiteA_2012-2020_Lv1_3.csv |
| citation: | http://doi.org/10.17592/001.2022041303 |
| reference: | Nishimura et al. (2023c) |
| **SIGMA-B** | |
| Level 1.1 | |
| data name: | Quality-controlled datasets of Automatic Weather Station (AWS) at SIGMA-B site from 2012 to 2020: Level: 1.1 |
| file name: | SIGMA_AWS_SiteB_2012-2020_Lv1_1.csv |
| citation: | http://doi.org/10.17592/001.2022041304 |
| reference: | Nishimura et al. (2023d) |
| Level 1.2 | |
| data name: | Quality-controlled datasets of Automatic Weather Station (AWS) at SIGMA-B site from 2012 to 2020: Level: 1.2 |
| file name: | SIGMA_AWS_SiteB_2012-2020_Lv1_2.csv |
| citation: | http://doi.org/10.17592/001.2022041305 |
| reference: | Nishimura et al. (2023e) |
| Level 1.3 | |
| data name: | Quality-controlled datasets of Automatic Weather Station (AWS) at SIGMA-B site from 2012 to 2020: Level: 1.3 |
| file name: | SIGMA_AWS_SiteB_2012-2020_Lv1_3.csv |
| citation: | http://doi.org/10.17592/001.2022041306 |
| reference: | Nishimura et al. (2023f) |


## 7. Summary and conclusion

This paper describes the in situ meteorological datasets from the SIGMA-A and SIGMA-B AWS
sites in northwest Greenland and details the QC methods used in preparing the datasets. At this time
when drastic environmental change is proceeding in the Arctic region, sound meteorological data and
QC methods are of ever-growing importance.
The QC method offered here consists of two basic steps. The first step, the initial control, masks
observations that are affected by mechanical malfunctions or local phenomena and is a pre-treatment
for the second QC step. This step uses simple statistics to set the range of permissible variation in
northwest Greenland for each observational parameter and flags erroneous records on the basis of that
variation range. The second QC step, the secondary control, masks erroneous observations based on
more stringent variation ranges as determined by the median and SD values of the full observation



record. The QC procedures offered here may be valuable for scientists developing their own QC
efforts.
We presented examples of time series of air temperature, snow height, PDD, atmospheric pressure,
snow temperature, surface albedos, and longwave radiation based on the resulting hourly
meteorological dataset for 2012–2020 in northwest Greenland. We also extracted information on
climatological cloudiness based on $LW_{net}$ data derived from these in situ ground observations. Our
primary findings are summarized in the following four points: (1) in the summers of 2015, 2016, 2019,
and 2020, high PDDs and low surface albedos were recorded at both SIGMA-A and SIGMA-B sites.
(2) Dramatic decreases in snow height occurred in 2015 at both AWS sites and in 2016, 2019, and
2020 at the SIGMA-B site. (3) Weather conditions in JJA were relatively variable in northwest
Greenland compared to the other seasons. (4) Clear-sky conditions typified the summers of 2015, 2019,
and 2020.
The datasets described here are archived in the open access Arctic Data archive System for all
scientific communities. We anticipate that they will not only aid in understanding and monitoring the
current climate in northwest Greenland but also contribute more broadly to the advancement of polar
climate studies.

**Author contribution**
All authors, excluding M. Nishimura, established the AWS systems and supported their
maintenance. In addition, M. Nishimura developed and carried out the QC procedures and analyzed
the observation data, TA designed and led the study project and provided technical support for the QC
procedures, M. Niwano conducted pre-treatments for the meteorological data record and constructed
a fundamental algorithm of the QC procedures, TY supported the field observations, especially
logistical support, and KF provided advice on interpreting the observational data. All authors
participated in the interpretation of results and gave final approval for publication.
**Competing interests**
The authors declare that they have no conflict of interest.
**Acknowledgments**
We recognize all members of the SIGMA project, the GRENE-Arctic Project in Greenland, and
the Arctic Challenge for Sustainability II (ArCS II) project. We also thank all of those who supported
the field observations. In particular, we thank Y. Iizuka (Hokkaido University), Y. Kurosaki (Hokkaido
University), and A. Tsushima (Chiba University) for taking part in the field activities at the SIGMA-
A site and establishing the AWS and Y. Komuro (National Institute of Polar Research) for technical

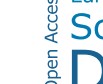

advice. This study was conducted as a part of the "Snow Impurity and Glacial Microbe effects on
abrupt warming in the Arctic (SIGMA)" Project supported by the Japan Society for the Promotion of
Science Grant-in-Aid for Scientific Research numbers JP23221004 and JP16H01772, the Global
Change Observation Mission-Climate (GCOM-C) research project of the Japan Aerospace
Exploration Agency, and ArCS II Program Grant Number JPMXD1420318865. For the use of
NunaGIS (http://en.nunagis.gl/) operated by Asiaq, Greenland Survey, in preparing Fig. 1, we
acknowledge the National Snow and Ice Data Center's QGreenland package (Moon et al., 2021). The
DEM data from Arctic DEMs were provided by the Polar Geospatial Center under NSF-OPP awards
1043681, 1559691, and 1542736.

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
