# Peer review of "Quality-controlled meteorological datasets from SIGMA 1"

_Earth System Science Data, 2023_

## Author Comment (AC2)

Response to Reviewer's comments (RC#1)

A response to the reviewer's comments is provided below. Reviewer comments are in black gothic type, and responses are in blue type.

11 May 2023
Referee Comment #1
Citation: https://doi.org/10.5194/essd-2023-116-RC1

This article is appropriate to support the publication of this data set. I was able to download the data and plot samples. I felt the accompanying metadata files and the readme files did a nice job in explaining the dataset. I felt this submission was of high quality and I would trust the dataset as useful.

Thank you for your comment. We appreciate your evaluation. We hope that this dataset will be published and will contribute to many studies.

---

## Author Comment (AC5)

Response to Editor's comments (EC#1)

Responses to the editor's comments is provided below. The comments are in black gothic type, and responses are in blue type.

26 Jun 2023
Editor Comment #1
Citation: https://doi.org/10.5194/essd-2023-116-EC1

I would like to take a moment and thank the reviewers for their work. There are a lot of good comments, which I would like to encourage the authors to respond to and then to prepare a revised submission.

In general, I believe that the manuscript would benefit from careful incorporation of the reviewers' comments.

Additionally, I would like to encourage the authors to:

- include the GC-net stations in figure 1, to show how this gap has been closed

We will revise as following the comment.

- clearly distinguish in the manuscript between measured and derived parameters

The observed values are summarized in Table 1, and the other parameters are derived parameters. If the distinction is made in the table, it would be more unclear, so we will revise the text to clearly state that "the observed values are summarized in Table 1 and the other parameters are derived parameters.

- consider publishing the processing code to increase confidence in the dataset.

As stated in our response to RC3's General comment, we will consider it positively and would prefer to publish it.

---

## Author Response (AR1)

Article reference: essd-2023-116
Title: Quality-controlled meteorological datasets from SIGMA automatic weather stations in northwest Greenland, 2012–2020
Author(s): Motoshi Nishimura et al.
MS type: Data description paper

Thank you for sending a decision letter that includes instruction of future revisions, and I appreciate giving a specific evaluation and valuable comments for improving our article from you and reviewers. Considering the comments, I have revised our manuscript as showing below. The given referee comments are written in gothic font, each comment's responses are in blue type, and correction details are in red type. Revised sentences and context are highlighted using track changes in MS Word in the text.

11 May 2023
Referee Comment #1
Citation: https://doi.org/10.5194/essd-2023-116-RC1

This article is appropriate to support the publication of this data set. I was able to download the data and plot samples. I felt the accompanying metadata files and the readme files did a nice job in explaining the dataset. I felt this submission was of high quality and I would trust the dataset as useful.

Thank you for your comment. We appreciate your evaluation. We hope that this dataset will be published and will contribute to many studies.

No specific correction was done.

18 May 2023

Referee Comment #2

Citation: https://doi.org/10.5194/essd-2023-116-RC2

**General:**

This paper presents approximately eight years of quality-controlled datasets of two automatic weather stations (AWS) on the ice in NW Greenland, one situated on the contiguous ice sheet and one on a detached coastal ice cap. This region is climatologically very interesting as well as rapidly changing, as described in numerous recent publications. These AWS data are highly valuable for process understanding, climate monitoring, and model evaluation/satellite validation, and deserve to be published. The data have been extensively quality controlled as described in this paper. The resulting dataset appears clean and robust and useful for users. My main problem is the non-concise and often unclear writing style in this paper, which makes the paper hard to digest and, more seriously, in places leads to confusion. Although it is a relatively minor remark, it will require a significant effort by the authors to remedy this.

Thank you for your comments; in accordance with the Major comment and the comments on Section 4.1, we have revised the text to reduce its length and improve readability. In addition, we will correct the unclear explanations in accordance with your comments.

**Major comments:**

I would like to encourage the authors to critically go through the MS text again to improve the readability and accuracy of the text. The writing can be more concise and precise. Some examples (not exhaustive) are listed below as minor comments. And in the process please aim for a shorter paper.

We will revise and improve the readability of this manuscript by following the reviewer's comments as

> *Section 4.1: I suggest listing all range values in Table 3 and not to repeat these in the text, to improve readability. Instead, for each correction it would be nice to mention the % data affected.*

and reducing the amount of text by deleting the text and figures in the sections on PDD (Fig. 5) and air temperature lapse rate (Fig. 7b).

The entire text has been reread, redundant parts have been reorganized, and unnecessary explanations have been deleted. I believe this work has improved the readability of the text.

We added the Tables summarized the ranges for each parameter as Table 3 in Sect. 4.1 and showing the percentage of unmasked data as Table 4 in Sect. 4.2. In Section 4.1, the description of the range

test written in Table 3 has been minimized, and the text has been modified as well. In section 4.2, the formula for the anomaly range of the anomaly test (Eq. 2.0.1) is noted at the beginning of the section and avoided being repeated in the text that follows.

In addition, with this correction, we found a new error in the lower limit of the range test for snow temperature, which is noted here and corrected.

Following parts were deleted.
- Figure 5 and the second paragraph of section 5.1 (both related to PDD)
- Figure 7b and the second paragraph of Section 5.2 (discussion on the rate of temperature decrease between the two locations)

Table 2 would be better placed at the very beginning or end of the text.

We have modified Table 2 to be placed at the front of the main text.

Table 2 has been moved to the beginning of Chapter 2. With this change, Table 1 was also moved to the same section to make the flow of the text more natural, and the title of Chapter 2 was changed.

This paper presents an observational dataset, so it is more logical to start the introduction with the history and importance of in situ observations in Greenland.

We thought I included an explanation in the second paragraph of the Introduction, but it seems it was insufficient, so we will add a few more sentences about the AWS network of GC-Net, K-transect, and PROMICE, citing their references and the position of SIGMA AWSs.

The position of SIGMA-AWS is clarified by adding an explanation of the history of the meteorological observation network in Greenland in the first part of Introduction. In addition, the first paragraph of the former version Introduction was deleted to make the flow of the text more natural.

Section 5 also discusses derived data, such as positive degree days, lapse rates but also average seasonal cycles, etc. Not sure such (admittedly basic) analysis has a place in a data journal.

Following the comment, we will delete the following parts
- Figure 5 and the second paragraph of section 5.1 (both related to PDD)
- Figure 7b and the second paragraph of Section 5.2 (discussion on the rate of temperature decrease between the two locations)

Those two parts (indicated above) were deleted.

**Minor comments:**

l. 17: an -> the

We will revise following the comment.

Corrected the relevant the part (L17 in the revised manuscript).

l. 25: ""snow height degradation", unclear, do you mean snow height decrease or snow metamorphism?

We will correct to "snow height decrease" since it means "snow height decrease".

Corrected the relevant the part (L28 in the revised manuscript).

l. 37: "however, the existing in situ meteorological data are insufficient for these purposes", unclear, do you mean that current observational coverage is insufficient? It is quite good in Greenland when compared to e.g., Antarctica.

"Insufficient" is an overstatement in some cases, so we will rephrase as "continuous accumulation of measured data will be more valuable".

No correction was done because the relevant part was deleted in the revision process.

l. 51: "analytical values of various numerical models", unclear, do you mean "output of numerical models"?

We will correct to "output of numerical models" as the reviewer's comment is correct.

Corrected the relevant the part (L53 in the revised manuscript).

l. 54: please explain 'sensor noise' and 'natural factors'.

Sensor noise mainly refers to a few watts of radiation that occurs at night. Currently, we use the terms "electric pulse," "electric noise," or "sensor noise" to refer to this error, but this was not appropriate. This phenomenon and its errors are generally referred to as "Zero Offset" (Behrens, 2021), so we will modify the terminology to use that terminology.

Natural factors include riming, ice accretion, snow accumulation on sensors, etc. Since this expression is indeed abstract, we will modify this part of the initial publication to add the following explanation: "natural factors"

-> "natural factors (e.g., riming, ice accretion, snow accumulation on sensors)"

Corrected the relevant the part (L55–57, 234–237, and 246 in the revised manuscript).

l. 56: please explain QC or better simply write out throughout.

Since QC was first mentioned here, we will add an explanation that QC is a process to improve the quality of data by removing outliers and modify the notation to Quality Control (QC).

Corrected the relevant the part (L58–59 in the revised manuscript).

l. 76: " It is considered" the fact that the surface consists of accumulating snow/firn proves that this is the accumulation area.

We will rephrase "It is considered to be" to "This site is".

Rephrased the relevant the part (L80 in the revised manuscript).

Fig. 1a: I suggest including the GC-Net stations as well.

We will consider adding GC-Net and K-transect observation sites.

The AWS locations of GC-Net and K-transect were added in Figure 1.

l. 78: "is supposed to be", unclear, was it intended to be at the equilibrium line, or is it thought to be there?

We intended that the SIGMA-B site is thought to be located at near the equilibrium line.

We will modify to mean that the SIGMA-B site is considered to be located at near the equilibrium line.

Corrected the relevant the part (L82 in the revised manuscript).

l. 80: " The surface condition at this site varies (see Fig. 2), and surface melting has occurred in warm years". Obviously, surface melting occurs at the equilibrium line. Did you perhaps mean "net ablation"?

Surface melting here was intended to mean "significant surface height decreasing," so "surface melting" will be changed to " significant surface height decreasing".

Corrected the relevant the part (L84–85 in the revised manuscript).

l. 97, Figure 2: mainmast -> main mast (also elsewhere in text).

We will correct following the comment.

Corrected the relevant the parts (many parts in the revised manuscript).

Figure 2: why is date given only in lower plots?

Since the surface condition of the SIGMA-B site varies greatly depending on the year and the time of year, we have included photographs of the different surface conditions at the site and the timing of each photograph in the figure as reference information. Specifically, in July in years with high temperatures, the entire snow layer may melt, exposing the bare ice surface, while in years with low temperatures, there may be almost no surface melting. This is a characteristic of the SIGMA-B site environment and is explained in the text.

No specific correction was done.

l. 113: cyclone battery?

Since "cyclone battery" was a proper noun, we will change the term to "lead-acid battery".

Corrected the relevant the part (L123 in the revised manuscript).

Table 1 caption typo: observaion -> observation

The submitted version spells it correctly, so it may be some kind of mistake on Referee's part.

In any case, it should be correct in the the revised manuscript.

No specific correction was done.

Table 1: accuracy of wind direction, unclear what is meant here.

We will check and correct the sensor specifications.

We checked the manual. Corrected the section on wind direction accuracy in Table 1. The accuracy of wind speed was also corrected, as it was not properly described.

Table 1: It appears that for the radiation measurement the sensitivity rather than the accuracy is listed?

We will check and correct the sensor specifications.

The manual was checked and the section on radiometer accuracy was corrected in Table 1.

l. 131: some, not all?

I thought it would be easier to explain $LW_{std}$ and others in the section explaining the QC process, and since they are not explained in the L131 section (section 3.2), I left them as some.

Since this is the intent, I will explain the intent and leave it as some instead of all.

No specific correction was done.

l. 133: "Because the vertical radiant flux against the inclined surface needed to accurately calculate the surface albedo and surface energy balance is affected by the sloping surface at the SIGMA-B site, we calculated the slope-corrected downward shortwave radiation (SWd_slope) from the corresponding observations using the correction method in Jonsell et al. (2003) and Hock and Holmgren (2005)." This sentence is unclear.

We will revise this sentence to be more concise and clearer.

We rephrased the relevant the part (L137–141 in the revised manuscript)

Table 2: in line 147, 'transmittance' is indicated by lowercase 't_r', in Table 2 we see an uppercase 'T_r' which is called 'transmissivity'. Are these the same things?

$T_r$ in Table 2 is a constant (0.881; SIGMA-A, 0.872; SIGMA-B) defined to explain the QC of shortwave and near-infrared radiation in secondary control, and $t_r$ in line 147 refers to the general

atmospheric transmission coefficient used in the $SW_{d\_slope}$ calculation, which is a different parameter. It may be confusing, so we will change the variable names Tr to $T_{rA}$ and $T_{rB}$, and $t_r$ is left unchanged to make it easier to distinguish them.

The name of the atmospheric transmittance variable in Table 2 have been modified so that they are general atmospheric transmittance variables ($t_r$) rather than site-specific constants for SIGMA-A and SIGMA-B. $T_r$ in the text has been modified to site-specific constant names, such as $T_{rA}$ and $T_{rB}$ (L339–340 in the revised manuscript).

Section 4.1: I suggest listing all range values in Table 3 and not to repeat these in the text, to improve readability. Instead, for each correction it would be nice to mention the % data affected.

We would like to reduce the amount of text by summarizing the range of the range test in a table or something. Also, it would be useful to know what percentage of the total data is masked, so we will add this information as a summary in a table or something.

The range test coverage is summarized in Table 3 and the percentage of masked data is summarized in Table 5.

l. 214: I do not understand this correction: why giving a clearly wrong measurement an arbitrary physical value?

Indeed, a negative wind direction is understood to mean that the sensor is making some kind of error, so we change the wind direction to be masked rather than set to 0 when the wind direction is negative. Corrected the relevant the part (L226–227 in the revised manuscript).

l. 224 and 236: 'electrical noise', what is this? Earlier you used 'sensor noise', is this the same?

Since they mean the same thing, we will unify them with "Zero Offset" (See Reviewer#1 L.54 comment response).
Corrected the relevant the part (L234–237, and 246 in the revised manuscript).

l. 226: why can alfa_sw and alfa_nir not be lower than 0.95 and 0.90? Or do you mean 'higher'?

It was a "higher" typo. We will correct it as such.
Corrected the relevant the part (L240 in the revised manuscript).

l. 232: Are these the conditions for which the data are flagged as erroneous? It seems to be the other way around.

It seems that the notation was not clear. These parts will be changed from
"$SW_d < SW_{TOA\_max}$," to "$SW_d > SW_{TOA\_max} \rightarrow SW_d = -9999$,"

and modify the text a little to match this modification.

In the revision process, above parts were decided to summarize in Table 3 and to delete the relevant part in the text.

l. 237-240: You give the data a physical value (zero), would it not be better to not do that unless for instance when SW_TOA < 0?

The negative emission that occurs when the solar zenith angle is greater than 90°, which means $SW_{\mathrm{TOA}}$ =0, is considered to be Zero Offset, so the negative value itself has no meaning, and there is no emission in such a case, so setting the value to 0 is not a problem.

Therefore, we will not modify this part.

No specific correction was done.

l. 249: the surface consists of snow or ice, so how can its temperature become positive?

Although the snow surface temperature is not higher than 0°C, a threshold of upward longwave radiation equivalent to a snow surface temperature of +10°C is set, taking into account the effect of radiation from AWS poles and other sources. This threshold does not imply that the snow surface temperature is positive.

Added explanation of above intention and changed the text in L258.

l. 299: Six hours of calm weather is not impossible, why this arbitrary value? Why not use the wind speed at the other AWS to check this?

Since there are no other AWS nearby, we cannot confirm the validity of this process.

Although the 6-hour period has an arbitrary, it is possible to be in a calm weather environment with infinitesimally small wind speeds for several hours, so the 6-hour threshold was set to avoid accidentally masking such a situation.

No specific correction was done.

l. 355: Why is wet snow treated differently at both sites?

The SIGMA-A site has a lower albedo limit of 0.3 because it is extremely unlikely that bare or dark ice will be exposed on the surface even if surface melting occurs, while the SIGMA-B site has a lower albedo limit of 0.1 because even dark ice may be exposed.

No specific correction was done.

Figure 4, 6, 8: Consider reducing symbol size.

We will revise as following the comment.

Corrected the relevant the part (Figures 4, 6, and 8).

09 Jun 2023
Referee Comment #3
Citation: https://doi.org/10.5194/essd-2023-116-RC3

Review of "Quality-controlled meteorological datasets from SIGMA automatic weather stations in northwest Greenland, 2012-2020" by M. Nishimura et al.
B. Vandecrux (bav@geus.dk)

The article describes the valuable meteorological data collected by two weather stations in northwest Greenland and the processing thereof. The article is clearly written, the key elements of the AWS systems are thoroughly described and the figures are of very good quality. I only have minor comments on the manuscript. It is great that this data is being published and distributed freely. Nevertheless I am concerned that many data users will wonder why there is only data up to 2020. Adding recent data will certainly increase reuse and citations. If this is not possible for some reason, then it should be stated clearly in the article. I am also strongly encouraging (and I think ESSD does as well) the publication of the scripts that are behind the data processing. This is key to making this dataset and article fully reproducible. After addressing these two points and the minor comments listed below, the article will be a great asset for ESSD.

Since July 2020, SIGMA-A observation data has been continuously showing erroneous and missing values due to some kind of malfunction. Due to the global pandemic of the COVID-19, field work has not been possible, and this situation continues to this day. Therefore, in this paper, we intend to publish the data set up to August 2020, when we can obtain the data reliably and when the mass balance year is well delimited.

As your comment, we are considering the possibility of releasing the processing code, as we believe it would benefit the scientific community. However, more time is needed to prepare for the release of the code, as more testing and code organization are needed.

We are working on those tasks now, but due to the large amount of work, we think it may be difficult to complete those tasks by the deadline of this revision, and we hope to complete the work and release the code by the time of publication.

**Comments on the article:**

- abstract: ESSD requires that the dataset DOI appears in the abstract. Please add the two DOIs of the two level 1.3 datasets.

We will add the dois of Level 1.3 datasets to abstract.

Corrected the relevant the part (L20–22 in the revised manuscript).

- l.22: "snow height increased" by how much? The use of "snow height" is misleading in the accumulation area. In many studies, snow more than one year old is not refered to as snow anymore but firn, so I first misunderstood this statement as the "annual snowfall is increasing". If the author do not distinguish snow and firn, then the total snow (+firn) thickness, and thickness change, are actually not measured. I recommend changing to "surface height" or "snow surface height". For Sigma-B, it would be nice to state clearly if it is standing on bare glacial ice. In that case "snow height" can be used.

We agree with the intention of the comment. The first installation was on bare ice in July 2012, but it is possible that refreezing ice (when it formed is unknown) is now forming above the bare ice surface in 2012. Therefore, "snow height" in the manuscript will be changed to "surface height" because a location higher than the surface height at the time of installation is not necessarily snow.

Rephrased the relevant the part (many parts in this manuscript), Tables 1 and 2, and Figures 3 and 4.

- l.24: "decrease" by how much? Again, for snow height, do you mean that the annual maximum snow height is decreasing or that the surface height is generally decreasing?

"decrease" means the surface is lowering. We will rephrase "snow height" to "surface height".

Corrected and rephrased the relevant the part (L26 in the revised manuscript).

- l.26: "notable snow height degradation" Not clear why it is notable or with regards to which normal it is a degradation. Please rephrase.

We will rephrase "notable" to "apparent".

Corrected the relevant the part (L28 in the revised manuscript).

- l.97: "mainmast" two words?

We will rephrase "mainmast" to "main mast".

Corrected the relevant the parts (many parts in the revised manuscript).

- l.123: "cm" line 104-105 you use m for instrument depth, now cm for height. Please be consistent. Potentially use only SI units.

We will change the "m" notation of the depth of snow temperature sensor installation to "cm" notation.

Corrected the relevant the part (L114–116 in the revised manuscript).

- table 1: It should be stated whether RH is provided with regards to water or with regards to ice (in subfreezing conditions). Some sensors do the conversion automatically, some don't. If it is with regards to water, then a corrected RH could be provided accounting for the different

saturation point in subfreezing conditions. Or at least potential correction methods should be listed.

Since this sensor calculates relative humidity based on the saturated water vapor pressure for liquid water, we will add a note to that in the table. The intent of this paper is to describe the QC method of the observed data and the observed values themselves, and it is our policy not to make any corrections or process the data including such a way that the intention of the implementer may intervene. Including further data processing methods in this paper would be redundant and would obscure the point of the discussion. We understand that accurate data analysis may require correction for shelter heating effects of air temperature and humidity in freezing environments, so we will discuss those treatments when we publish such a paper. However, We will revise the text to add a note to that effect, for alerting readers to this issue.

Added a paragraph with explanations in Sect.3.2 and a note in Table 1.

- Section 3: I am missing a discussion of the sensors' known limitations, it could be either included under the AWS system description subsections or in a section of its own at the end of the manuscript. It should estimate how often those problems my occur and point at potential way to remidiate them. Some of these limitations are:

It may be necessary to describe the errors that the observed values contain, and we will add a subsection in Chapter 3 to explain this. The response to the correction and limitation of individual observations is described in detail below.

However, as noted in the response to the comment on Table 1, the intention of this paper is to publish the observed values themselves, without any correction or data processing that might involve the intervention of the implementer's intention. Therefore, we will note that the data published in this paper possibly contain some errors only and will not conduct any additional analysis or corrections that would show the corrected values.

Added a paragraph with explanations in Sect.3.2 that some data may require correction.

- RH sensor clogging up with rime (https://doi.org/10.1007/s10546-004-7955-y)

The temperature and humidity sensors used at both sites may be affected by icing and riming as you have indicated. We will revise the text to cite this paper and add a note to that effect, for alerting readers to this issue.

Added a paragraph with alert explanations in Sect.3.2 that RH data may require correction.

- Unventilated thermometer overheating in low wind and clear sky conditions.

As noted in the response to the comments on Table 1, this paper does not include any correction or

data processing that might include the possibility of intervening intentions of the implementer, and the intention is to publish the observed values themselves, so I will not discuss such issues. However, We will note the shelter heating effect, which has been pointed out in many previous studies, in Chapter 3.

Added a paragraph with alert explanations in Sect.3.2 that the possibility of positive bias due to the shelter heating.

- Radiation sensors and anemometers being shadowed/sheltered by the station mast (https://doi.org/10.1029/2010JD015507)

At least radiation sensors of SIGMA-A is placed far enough away from the AWS main mast, and the pole of the radiometric sensor is placed in such a way that it does not affect the sensor. So we think that the station mast has almost no influence to those sensors. If SIGMA-B is affected, it would be by the shadow of the satellite communication antenna mounted at the top of the main mast of AWS. I cannot make a quantitative assessment of the presence or absence of this effect, but a detailed review of the hourly data showed that the effect was not pronounced. According to this, it is highly unlikely that the antenna's shadow is affecting the radiation, and if it is, it is likely to be slight. Therefore, we think no specific treatment is required. Nevertheless, your point is a valid one, and I will add a brief summary of the above explanation to Chapter 3.

Added a paragraph with explanations in Sect.3.2 that although there is a possibility of being made shadow on the radiation sensors by the AWS mast, the effect did not confirm by checking the dataset.

- l.218: if RH is given with regards to ice, then supersaturation is not uncommon on the ice sheet up to ~110% and this filter may be too strict. If RH is given with regards to water then values >100 are unlikely.

Since the humidity sensor measurement is based on relative humidity relative to liquid water, we would leave the upper threshold at 100%.

Added a paragraph with explanations in Sect.3.2 and a note in Table 1.

- l.226: "lower" higher?

The indication is correct, we will correct it to "higher".

Rephrased the relevant the part (L240 in this manuscript).

- l.243: Please avoid this use of brackets in equations to indicate interchangeable variables. Brackets have a defined meaning in equations. Either spell out two equations or use a subscript "i" in the equation and define it in the text like: "i being either u or d"

We will correct as per the comment with some subscripts.

We decided to use χ as a subscript and added a note to L197–198 that χ indicates the downward or upward direction of the radiation. The specific change in the text in L247–249, and 344 in the revised manuscript.

Although the comments were only for the upward and downward emissions, Procedure 1.4.2 also contained parentheses, so the variable was changed and an explanation for the variable was added to the L253–255.

- Section 4.1.5: Do you use the same filters for "sensor_height" as for "sh"? It should be mentioned in the text.

Since the sensor height is calculated after the QC of the snow height was completed, we do not set any no filter for the sensor height. I will add an explanation to the text about this.

The explanation of above was added in the last part of Sect. 4.2.6.

- l.317: "weak electric pulse" where does that pulse come from, why is it weak and how does this relates to the radiation measurements?

"weak electric pulse" mainly refers to a few watts of radiation that occurs at night. The radiation amount is an observation error caused by the specifications of the instrument, and the error is caused by the slight temperature difference between the two detectors (inside of the dome shelter and sensor body), which occurs when there is a large temperature difference between the outside air temperature and the temperature inside the sensor body.

This radiometric error may cause the shortwave radiation to be recorded as an observed value at night. However, since the value is an observation error, the observed value may be different from the original radiation balance.

The explanation of above was added in the first paragraph of Sect. 4.1.3.

- l.331: same comment as line 243

We will correct as per the comment with some subscripts.

Same as the response for the L243 comment, we corrected the relevant the part.

- l.409: since the snow temperature sensors' installation depths were given in meter, I misunderstood the "-1" as meter. Please be consistent with the units.

We will correct the notation of depth for snow temperature sensor installation to "cm".

No specific correction for this comment was done, because the notation of the unit of initial sensor installation depth has been corrected.

- l.445: please give mean annual PDD and its standard deviation to support this statement.

In accordance with RC2, we are going to reduce the text, so we will delete the part about the analysis of the PDD. Therefore, the relevant part of this comment will also be deleted, so I will not respond to it.

No specific correction for this comment was done.

-l.454: Is there any net ablation years? Does the station allow to measure the ice ablation? Is there any measurement (e.g. stakes) of the ice ablation? Please elaborate on this.

Since no ice thickness changes or stake observations were made, it is not possible to discuss the mass balance. Since this discussion is based on observations at AWS, the discussion is based on meteorological observation data.

Incidentally, Sugiyama et al. (2021) reported the result of stake observations of the SMB for the years 2012/13-2018/19. The result showed that the SMB at the same elevation zone, the clearly negative SMB year is 2014/15. 2015/16 and 2018/19 are ±0, and the rest are POSITIVE. However, since this is not an observation at the SIGMA-B site and we did not observe it at the same elevation as the SIGMA-B site, we do not know if its SMB is the same at the SIGMA-B site. This verification is beyond the scope of this paper and will not be done in this paper.

No specific correction was done.

- l.472: shouldn't the lapse rates be negative?

The point is correct, but I will delete this section and will skip responding to your comment.

No specific correction was done.

**Comments on the data files:**

- Commercial formats like Microsoft Word should be avoided. Please replace by a text file.

The temporary Dropbox data link may have included MS word files, but the official dataset data and doc do not include MS word, so please check the doi link page.

No specific correction was done.

- The station coordinates (potentially through time?) and a table giving the meaning of each variable (as they are named in the data files) should be provided at least in the readme file, or even better: in separate, machine-readable files (e.g. csv, tsv).

We will change information published as pdf files to text format, etc.

The station coordinates, and easy information of each variables were added in the read me files.

Details for each variable were not written in the read me file, because it makes the readme file redundant and decreases a readability, and they are also included in the body of the paper.

- The date format used is non-standard. ESSD encourages ISO 8601 (or alike). Please specify if time stamp is local time or UTC.

We will correct the time data format.

We will update the dataset again after the peer review process is over, and I will correct the timestamps at that time.

26 Jun 2023

Editor Comment #1

Citation: https://doi.org/10.5194/essd-2023-116-EC1

I would like to take a moment and thank the reviewers for their work. There are a lot of good comments, which I would like to encourage the authors to respond to and then to prepare a revised submission.

In general, I believe that the manuscript would benefit from careful incorporation of the reviewers' comments.

Additionally, I would like to encourage the authors to:

- include the GC-net stations in figure 1, to show how this gap has been closed

We will revise as following the comment.

The AWS locations of GC-Net and K-transect was added in Figure 1.

- clearly distinguish in the manuscript between measured and derived parameters

The observed values are summarized in Table 1, and the other parameters are derived parameters. If the distinction is made in the table, it would be more unclear, so we will revise the text to clearly state that "the observed values are summarized in Table 1 and the other parameters are derived parameters. Added explanation of above intention in L75.

- consider publishing the processing code to increase confidence in the dataset.

As stated in our response to RC3's General comment, we will consider it positively and would prefer to publish it.

The program is now available as an asset of this paper.

---

## Referee Report (RR1)

Review of ESSD manuscript version essd-2023-116-ATC1.pdf.

Quality-controlled meteorological datasets from SIGMA 2 automatic weather stations in northwest Greenland, 2012– 3 2020

*by Nishimura and others*

**General**

As all previous reviewers, I want to honor the authors for their high efforts in maintaining weather stations in a harsh environment and for providing the data to the scientific community. I also acknowledge their attempts to increase the quality of the dataset by removing erroneous measurement values and discussing the time series.

While I have the impression that the authors spent substantial work in making use of the already available reviewers' and editor's comments, I found a few points that might be considered/revised before the paper is being published.

Best wishes

Wolfgang Gurgiser

**Comments:**

On section 3.2

- Do you correct SWd (incoming shortwave radiation?) at SIGMA-B for the 4° slope angle? If yes, does this mean that your station/CNR4 is not mounted horizontally but in parallel to the slope?
- I'm not sure if you use the corrected data (SWd/u_slope) for calculating the surface albedo because in equation ix, there is no "slope" subscript
- Might the lower sensors of the CNR4 at SIGMA-B be markedly influenced by the station, also by the solar panels? If yes, you should also mention this influence at the end of the section

By line(s), figures or tables

Line 24: "A proxy of cloud formation frequency" – I would just write a "A proxy of cloudiness" because incoming longwave radiation just indicates if clouds are there or not

Lines 24-29: While I find a comparison per-se interesting, it's in my opinion not necessarily needed in the abstract of this paper. If you keep it, I would recommend revisions because in its current form, it's hard to understand.

Line 65: Maybe change to "…support the evaluation and development of numerical models"

Line 66: I suggest to delete the second part of the sentence starting with "and…"

Line 69: I suggest to change the text as follows "…or technical issues (e.g. Zero Offsets, faulty sensors)"

Line 70: "quality" instead of "accuracy"?

Line 71: I would remove "which is a process …" because e.g. detecting a drift in a sensor or detecting unrealistic constant values over time would for me also be part of quality control

Line 88: "…and the sensor…"

Lines 88-90: Please split the sentence in two.

Line 93: "…an accumulation area of the ice sheet…"

Line 95: "…located near…"

Line 98: "…vary"

Line 98-99: e.g. "…surface lowering…" (surface height decreasing is not a good expression)

Line 99: "…is located…"

Line 135: "…below…"

Line 206: "… and when the…"

Line 244: Maybe change to "…of simple statistics based on maximum, minimum, and mean values derived from …"

Lines 249-251: Sorry, but I don't get the meaning of the added text.

Line 270: "...from the entire (???) observation…"

Equation 1.4.3: I wonder if there could be conditions (e.g. fog) where LWd and LWu could be very similar in your environment? Then, maybe not all time steps with LWd very close to LWu should be flagged.

Line 359: "…of correct values…"

Table 5: % SWu is higher in Level 1.3 than in Level 1.2? This might be an error.

Line 394: "…is shown…"

Line 398: "…calculated only from the…"

Line 403-404: Wouldn't riming or icing cause shading of the sensor and thus, rather an underestimation of the SW values?

Line 421-422: Do you mean "…that passed the second control"?

Table 4: Might it be that you do not show the threshold values but the min/max values that you use to determine the threshold values? In other words, is e.g. the threshold of wind speed at U2A 25.5+15 m/s? If yes, I would add the real thresholds in the table.

Lines 467-468: What differences between LWstd and the measured LW values is allowed for not flagging a value?

Equation 2.7.1: Please add the unit (cm?).

Line 528: This text sounds very subjective, maybe you can explain more about potential valid records that would be masked.

Line 536: "The maximum was slightly positive at SIGMA-A site"- which maximum? In case you refer to the +7.2°C, I would not write "slightly positive" but just provide a number.

Fig. 4: In the legend, please replace snow height with surface height.

Fig. 7: The subplot showing the lapse rate is not described in the figure caption any more. Please correct.

Line 630: I'm not familiar with "occurrence frequency" but rather "frequency distribution" – you may check again.

Lines 639-642: This text could be shortened.

Line 653: "…indicates a higher frequency of clear…."

Line 655: Isn't the sentence starting in this line replicating the information in lines 649-650?

Lines 689-691: Please revise the description of (1)

Line 691: Maybe you find a better description instead of "dramatic"

Line 693: Maybe change to "(3) Observed atmospheric conditions in JJA…"

Line 694: "(4) Frequent clear-sky…" or even "(4) Low cloudiness" which might be a more accurate description than "clear-sky" throughout the paper

---

## Author Response (AR2)

Article reference: essd-2023-116

Title: Quality-controlled meteorological datasets from SIGMA automatic weather stations in northwest Greenland, 2012–2020

Author(s): Motoshi Nishimura et al.

MS type: Data description paper

Thank you for sending the specific and valuable revision comment on the revised manuscript. Considering the comments, we have revised our manuscript as shown below. The given referee comments are written in gothic font. Each comment's responses and correction details are in blue type. Revised sentences and context are highlighted using the text's track changes in MS Word.

We also added our thanks to the reviewers and editors for their help in the peer review process.

20 Sep. 2023

Referee Comment #4

General

As all previous reviewers, I want to honor the authors for their high efforts in maintaining weather stations in a harsh environment and for providing the data to the scientific community. I also acknowledge their attempts to increase the quality of the dataset by removing erroneous measurement values and discussing the time series.

While I have the impression that the authors spent substantial work in making use of the already available reviewers' and editor's comments, I found a few points that might be considered/revised before the paper is being published.

Best wishes

Wolfgang Gurgiser

**Comments:**

On section 3.2

• Do you correct SWd (incoming shortwave radiation?) at SIGMA-B for the 4° slope angle? If yes, does this mean that your station/CNR4 is not mounted horizontally but in parallel to the slope?

$SW_d$ is the observed value of the horizontally installed shortwave radiation sensor (CNR4) itself. $SW_{d\_slope}$ is the physical quantity obtained by correcting the observed $SW_d$ as the amount of radiation for a slope inclined by 4°.

• I'm not sure if you use the corrected data (SWd/u_slope) for calculating the surface albedo because in equation ix, there is no "slope" subscript

Since there is no need to correct for slope at the SIGMA-A site, we use this general notation. However, because of the possibility of such misunderstanding, we added the explanation in L170 in the revised manuscript that "$SW_{d\_slope}$ is used for $SW_d$ when calculating $a_{sw}$ at the SIGMA-B site".

• Might the lower sensors of the CNR4 at SIGMA-B be markedly influenced by the station, also by the solar panels? If yes, you should also mention this influence at the end of the section

We did not identify the significant influence of AWS including solar panels during the process of examining the QC procedure. However, the possibility of such an effect cannot be completely ruled out, and an explanation is added at the end of section 3.2.

**By line(s), figures or tables**

Line 24: "A proxy of cloud formation frequency" – I would just write a "A proxy of cloudiness" because incoming longwave radiation just indicates if clouds are there or not

We have corrected it as you indicated.

The corrected part is L24 in the revised manuscript.

Lines 24-29: While I find a comparison per-se interesting, it's in my opinion not necessarily needed in the abstract of this paper. If you keep it, I would recommend revisions because in its current form, it's hard to understand.

The policy is to maintain a comparison of the climates of two sites that are relatively close to Northwest Greenland, and the text has been revised to make it clearer.

The revised part is L24–29 in the revised manuscript.

Line 65: Maybe change to "…support the evaluation and development of numerical models"

We have corrected it as you indicated.

The corrected part is L55 in the revised manuscript.

Line 66: I suggest to delete the second part of the sentence starting with "and…"

We have deleted the relevant part as you indicated.

The revised part is L55 in the revised manuscript.

Line 69: I suggest to change the text as follows "…or technical issues (e.g. Zero Offsets, faulty sensors)"

We have revised it as you indicated.

The revised part is L58–59 in the revised manuscript.

Line 70: "quality" instead of "accuracy"?

We have corrected it as you indicated.

The corrected part is L60 in the revised manuscript.

Line 71: I would remove "which is a process …" because e.g. detecting a drift in a sensor or detecting unrealistic constant values over time would for me also be part of quality control

The points you indicated are also intended to show that what QC is intended to do in this paper. Therefore, if this sentence is deleted, the definition of QC in this paper will become ambiguous. Although we can understand the intent of the comment, we will not change this section for the reasons stated above.

Line 88: "…and the sensor…"

We have corrected it as you indicated.

The corrected part is L78 in the revised manuscript.

Lines 88-90: Please split the sentence in two.

We have corrected it as you indicated.

The corrected part is L79 in the revised manuscript.

Line 93: "…an accumulation area of the ice sheet…"

We have corrected it as you indicated.

The corrected part is L83 in the revised manuscript.

Line 95: "…located near…"

We have corrected it as you indicated.

The corrected part is L85 in the revised manuscript.

Line 98: "…vary"

We are sorry, but we did not understand the intent of this comment. Grammatically, the verb in this passage is "varies", so we did not make the correction.

If you have some other intents, please let me know.

Line 98-99: e.g. "…surface lowering…" (surface height decreasing is not a good expression)

We have corrected it as you indicated.

The corrected part is L88 in the revised manuscript.

Line 99: "…is located…"

We have corrected it as you indicated.

The corrected part is L89 in the revised manuscript.

Line 135: "…below…"

We have corrected it as you indicated.

The corrected part is L119 in the revised manuscript.

Line 206: "… and when the…"

We have corrected it as you indicated.

The corrected part is L182 in the revised manuscript.

Line 244: Maybe change to "…of simple statistics based on maximum, minimum, and mean values derived from …"

We have corrected it as you indicated.

The corrected part is L218 in the revised manuscript.

Lines 249-251: Sorry, but I don't get the meaning of the added text.

We revised the text in the relevant section.

The revised part is L L224–226 in the revised manuscript.

Line 270: "...from the entire (???) observation…"

As you pointed out, we are referring to the entire period. It was indeed a confusing expression, so we have rewritten it in a concise manner as you suggested.

The corrected part is L236 in the revised manuscript.

Equation 1.4.3: I wonder if there could be conditions (e.g. fog) where LWd and LWu could be very similar in your environment? Then, maybe not all time steps with LWd very close to LWu should be flagged.

As you point out, $LW_{net}$ could be smaller in weather conditions such as fog. And certainly in such cases that data should not be masked.

We tried to determine if there was fog or frost using temperature and humidity data. For example, when the humidity is very high (e.g., 90-100%), if the temperature is negative, it is frost, if positive, it is fog, and so on. However, fog may occur in a freezing condition, the separation of those conditions is difficult, and the optimal algorithm cannot be determined. Therefore, we were led to the conclusion

that more careful consideration is needed to determine an effective masking algorithm.
Therefore, this is an issue for future improvement, and for now, the algorithm will not be changed with the policy of broadly masking suspect data. Thank you for your helpful comments.

Line 359: "…of correct values…"
We have corrected it as you indicated.
The corrected part is L305 in the revised manuscript.

Table 5: % SWu is higher in Level 1.3 than in Level 1.2? This might be an error.
Thank you for pointing this out. I checked the algorithm and found a minor bug. We have fixed the bug and replaced Table 5.
In the process of making this correction, we also found an error in the text of the "anonymous test" described in section 4.2.3, so we have corrected the part in L343–344 in the revised manuscript as well.

Line 394: "…is shown…"
We have corrected it as you indicated.
The corrected part is L338 in the revised manuscript.

Line 398: "…calculated only from the…"
We have corrected it as you indicated.
The corrected part is L342 in the revised manuscript.

Line 403-404: Wouldn't riming or icing cause shading of the sensor and thus, rather an underestimation of the SW values?
We believe it depends on the intensity of liming and icing. If a thin ice film is formed on the dome, it may cause multiple reflections within the dome, resulting in the overestimation of radiation. If the thick one is formed, we identified that using $LW_{net}$ (Equation 1.4.3). In this case, the radiation data have been masked.

Line 421-422: Do you mean "…that passed the second control"?
That is exactly how I understand it. It seems that it was difficult to understand, so the text has been revised.
The revised part is L358 in the revised manuscript.

Table 4: Might it be that you do not show the threshold values but the min/max values that

you use to determine the threshold values? In other words, is e.g. the threshold of wind speed at U2A 25.5+15 m/s? If yes, I would add the real thresholds in the table.

You are correct, the values used only to determine the Range test expression thread are listed in this table. The threshold values used in the Range test themselves are summarized in Table 3, so we do not list them in this table.

We have corrected the caption of the table to avoid possible misinterpretation.

The corrected part is L292 in the revised manuscript.

Lines 467-468: What differences between LWstd and the measured LW values is allowed for not flagging a value?

The coefficients used in the Anomaly test (see Eq. 2.0.1) are listed in the second paragraph of Section 4.2.5. This coefficient will give you an idea of the Anomaly range. Also, specific median and standard deviation values are not included because it would be redundant to include them in the paper. If necessary, they can be found in the published program. By adding a note in the revised manuscript (L303–304) that those statistical values can be referenced in the program as needed, we thought that the intent of the anomaly test for LW could be understood by this explanation.

Equation 2.7.1: Please add the unit (cm?).

All equations describing QC procedures, including Eq. 2.7.1, omit the units. Therefore, we have added the units in the text, not in the equations.

We also thought it would be better to use cm as the unit of *st_depth* in Table 2, so we have revised that part as well.

The corrected parts are Table 2 and L424 in the revised manuscript.

Line 528: This text sounds very subjective, maybe you can explain more about potential valid records that would be masked.

It is understandable that this process involves subjectivity. However, in the case of a data where there is very little erroneous data set such as atmospheric pressure, it is difficult to determine the ranges of the range and anomaly tests based on statistical values, and this is the reason why this process was conducted.

It is clear from the Level 1.2 dataset that the data that seem to be anomalies clearly deviate from the fluctuations of the atmospheric pressure. We checked the data after processing and confirmed that the erroneous data were masked appropriately and that other correct values were not masked. We have added an explanation to the extent that it is not redundant, just to be sure.

The revised part is L441–443 in the revised manuscript.

Line 536: "The maximum was slightly positive at SIGMA-A site"- which maximum? In case you refer to the +7.2°C, I would not write "slightly positive" but just provide a number.

Thank you for pointing this out. I certainly did not explain it well enough. This discussion part was regarding the annual maximum monthly mean temperature. Including other parts of the text besides the part you pointed out, we have revised the text to make the interpretation of the data more accurate. The revised part is L449–454 in the revised manuscript.

Fig. 4: In the legend, please replace snow height with surface height.

They should have been corrected to surface height in the last revision. Therefore, this section will not be revised.

Fig. 7: The subplot showing the lapse rate is not described in the figure caption any more. Please correct.

The part of Fig. 7 showing the rate of temperature diminishing was removed in the previous revision. The caption has been removed accordingly, and the explanation related to the section has been removed from the caption as well.

Line 630: I'm not familiar with "occurrence frequency" but rather "frequency distribution" – you may check again.

We thought the frequency distribution was more prenatal as you pointed out, so I modified it accordingly.
The modified part is L522 in the revised manuscript.

Lines 639-642: This text could be shortened.

We have revised the sentences following the comment.
The revised part is L532–535 in the revised manuscript.

Line 653: "…indicates a higher frequency of clear…."

We have corrected it as you indicated.
The corrected part is L546–547 in the revised manuscript.

Line 655: Isn't the sentence starting in this line replicating the information in lines 649-650?

You are correct. Since the repetitive parts cannot be deleted due to context, we have reorganized the information and revised the text to avoid redundancy.
The revised part is L548–551 in the revised manuscript.

Lines 689-691: Please revise the description of (1)

The relevant part has been rewritten in a concise manner.

The revised part is L583–585 in the revised manuscript.

Line 691: Maybe you find a better description instead of "dramatic"

We have rewritten "dramatic" to "apparent".

The revised part is L586 in the revised manuscript.

Line 693: Maybe change to "(3) Observed atmospheric conditions in JJA…"

We have corrected it as you indicated.

The corrected part is L587 in the revised manuscript.

Line 694: "(4) Frequent clear-sky…" or even "(4) Low cloudiness" which might be a more accurate description than "clear-sky" throughout the paper

We think your point is valid. We have revised the part you pointed out and the abstract to describe "frequency of clear skies" instead of "clear-sky".

The corrected part is L26 and 588 in the revised manuscript.

---

## Author Response (AR3)

Article reference: essd-2023-116

Title: Quality-controlled meteorological datasets from SIGMA automatic weather stations in northwest Greenland, 2012–2020

Author(s): Motoshi Nishimura et al.

MS type: Data description paper

Thank you for revision comments on the revised manuscript. Some of them were simple careless mistakes, for which we apologize. We have revised our manuscript as shown below. The given comments are written in gothic font. Each comment's responses and correction details are in blue type. Revised sentences and context are highlighted using the text's track changes in MS Word.

During this revision work, the following corrections other than those pointed out were made.

- Correction of misalignment of figure numbers
- Deletion of the description of Figure 5 in the text

We have deleted Fig. 5, showing temporal variation of PDD, in the first revision, but realized that we forgot to correct the figure number at that time. In this revision, the point and the description of the citation of the figure in the text were corrected. In addition, the deleted description of Fig. 5 remained in the text, which was also deleted in this revision. The specific revised parts are L485 and L507 in the main text.

20 Sep. 2023

Editor Comment

I would like to thank the authors for the revisions. I noticed that corrections to the manuscript introduced a few language errors which should be corrected.

All line numbers refer to the track-change manuscript.

I am suggesting the following technical changes to the manuscript language without changing scientific content, but this should be checked by the authors.

L28-29: Therefore, it appears that weather condition differences led to the apparent ...

We have corrected it as you indicated.

L184-5: "In addition, reflection and shielding of scattered radiation due to the AWS including solar panels may ..."

We have corrected it as you indicated.

L303-304: " Those statistical values and multipliers can be found in the QC program (archived at Ref) .

We have corrected it as you indicated.

In adding the source code reference, we have also added the program citation information in the Reference section.

L343: "If the record is within its

We have corrected it as you indicated.

L358: "datasets that passed"

We have corrected it as you indicated.

L441-443: This threshold value of 20 hPa is based on the assumption that a 20 hPa pressure jump is unlikely to occur in a few hours. This procedure was successful in only masking erroneous data of both sites.

We have corrected it as you indicated.

L620: Tobias Gerken

We have corrected it as you indicated.